# Measurement instruments for sexual identity minority stress in adults: A scoping review

**Maria Misevic-Kallenbach**[1]*, **Susann Conrad**[2], **Simone Freitag**[3], **Anja Jacobs**[4], **Jacqueline Schirm**[5], **Madlen Sixtensson**[6], **Petra Warschburger**[1]

1 Department of Psychology, University of Potsdam, Potsdam, Germany, 2 Independent Researcher, Berlin, Germany, 3 Institute of Nursing Care Science and Interprofessional Learning, University Medicine Greifswald, Greifswald, Germany, 4 The German Cancer Society, Berlin, Germany, 5 Independent Researcher, Schöneiche, Germany, 6 Independent Researcher, Potsdam, Germany

* misevickallenbach@uni-potsdam.de

## Abstract

Lesbian, gay, bisexual, and queer (LGBQ) populations face disproportionate health inequities shaped by minority stress processes. To enable consistent and comparable assessment, this scoping review maps measurement instruments for sexual-identity minority stress. Following PRISMA guidelines, the review involved an extensive literature search across Medline, Embase, PsycInfo, PSYNDEXplus, PTSDpub, Web of Science, the GASP Measures Database and GESIS Open Access Repository for Measurement Instruments (last updated in February 2025), resulting in 39,642 references. Selection criteria focused on instruments measuring minority stress in form of stigmatization, discrimination, victimization, internalized homophobia, expectations of rejection, identity concealment, as well as positive aspects of minority identity among adult LGBQ populations. Gender-identity-related stressors were outside the scope. Only articles in English and German were considered. No restrictions were applied with regard to year of publication or publication medium. A total of 105 instruments with 152 references met the criteria and were included for detailed analysis. The analysis identified a broad spectrum of instruments, predominantly targeting internalized homonegativity and stigmatization. The review also uncovered the risk of jingle-jangle fallacy, attributable to inconsistent naming and definition of constructs across instruments. An underrepresentation of instruments for lesbian, bisexual and non-monosexual populations, and a trend towards instruments emerging from grey literature sources was observed. This scoping review demonstrates a rich diversity in instruments measuring sexual minority stress but reveals gaps in gender inclusivity. The findings emphasize the importance of expanding the scope to include intersectionality and diverse cultural contexts. PROSPERO registration number: CRD42021257995.

provided the original author and source are credited.

**Data availability statement:** All relevant data are within the manuscript and its Supporting Information files.

**Funding:** The author(s) received no specific funding for this work.

**Competing interests:** The authors have declared that no competing interests exist.

## Introduction

The pursuit of equal rights by Lesbian, Gay, Bisexual, and Queer (LGBQ) activists has raised awareness of health disparities within this community, initially emphasized on the sexual health of gay men due to the HIV/AIDS epidemic [1]. However, such a focus potentially neglects the multifaceted experiences of LGBQ individuals, extending beyond sexual health inequalities. LGBQ individuals face unique challenges such as structural stigma [2,3], barriers to care [4,5], and discrimination and violence because of their sexual orientation [6]. Current research increasingly reveals these challenges and distinct health concerns within this group, including higher rates of depression, anxiety, substance abuse disorders, and suicidal tendencies compared to their heterosexual counterparts [7–10].

### The minority stress model

Central to understanding these challenges is Meyer's minority stress model [11,12],which posits that the health inequalities observed among LGBQ individuals are a result of social, rather than intrinsic factors related to their sexual identity. This model (see Fig 1) identifies both distal (external) stressors and proximal (internal) specific to the LGBQ population. The model also notes that the degree to which one's identity is intertwined with the LGBQ community can mediate the intensity of these experiences.

**Distal stress factors.** Meyer conceptualizes distal stressors as objective stressors aimed at minority populations, including *discrimination* and *victimization* which constitutes key external factors that can detrimentally impact health [11,12]. These experiences are also termed prejudice events [14] or enacted stigma [15]. Across designs and settings, evidence shows that enacted stigma is common, spans over a continuum from microaggressions in the form of slurs, derogatory remarks to overt social exclusion, ostracism, harassment and violence, and constrains everyday participation in school, community, and service environments [16]. Its health impact is consistent across contexts: cohort data link enacted stigma to sexual risk behaviors [17], while evidence from China indicates that higher levels of enacted stigma, assessed with the China MSM Stigma Scale [18] tailored specifically to this cultural context, are positively associated with adverse mental health in this population [19].

**Proximal stress factors.** Proximal processes can exacerbate these stressors, such as the anticipation of negative events and the internalization of LGBQ-hostile ideologies [11]. Meyer conceptualizes the latter as *internalized homophobia* [11,14], also termed internalized homonegativity [20] or internalized heterosexism [21], meaning the turning of negative social attitudes toward the self. Irrespective of the degree of outness or self-acceptance, early socialization and continued exposure to anti-LGBQ norms can shape self-appraisals and internal conflict, with downstream effects on self-regard [22].

Internalized homophobia has been examined extensively with respect to health correlates and assessment [20,21,23]. Higher internalized homonegativity is consistently associated with poorer mental health, including increased depressive

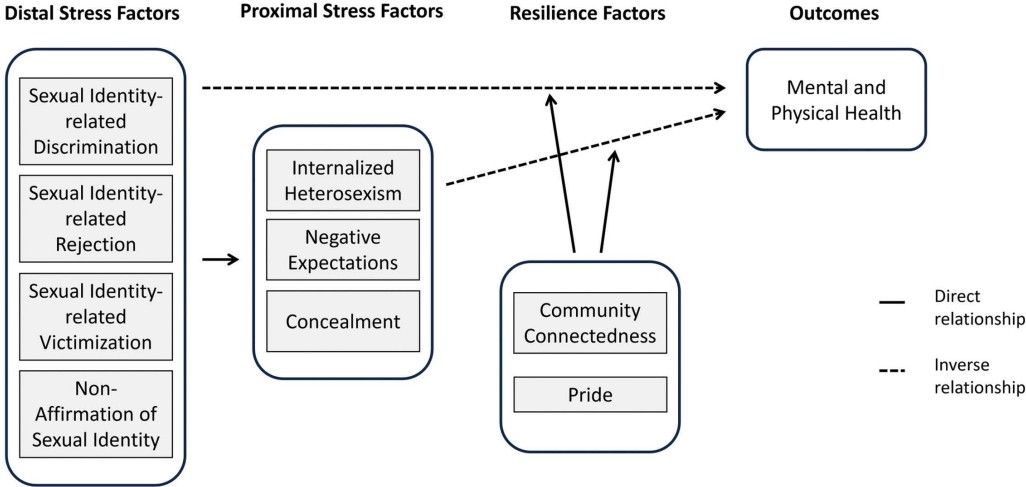

**Fig 1. The minority stress model [11], adapted from Testa et al. [13].**

symptoms and reduced life satisfaction [22,24–27]. Nevertheless, definitions and operationalizations remain heterogeneous [22,24,28–31]. One early measure of internalized homophobia is the Nungesser Homosexual Attitudes Instrument (NHAI) [32]. Its widespread use was questioned considering societal changes since its publication in 1983 [22]. Further, researchers have differed on construct breadth when measuring internalized homophobia, with some measures conflating internalization with outness and community connectedness [28,29,33], or embedded indicators of suicidality and depression like in the NHAI [29,32]. In Meyer's minority stress model, internalized homophobia, concealment/outness, and community connectedness are conceptually distinct [24]. Comparative analyses indicate that scales which blend internalization with distress yield stronger associations with health than scales that treat internalization orthogonally to distress [23], underscoring the need for conceptually precise instruments.

The anticipation of distal stressors is another proximal factor in the minority stress model, referred to as *expectations of rejection* [11]. The tendency to anticipate rejection and to read social cues, especially stigma-relevant ones, as rejection actions has been termed rejection sensitivity by Pachankis et al. [34,35]. Yet, others distinguish rejection sensitivity from expectations of rejection, defining the former as the joint presence of a cognitive appraisal of the likelihood of rejection and an affective response marked by concern or anxiety about it [36]. In a community sample of gay men using the gay-related rejection sensitivity scale showed that internalized homophobia mediated the link between parental rejection and rejection sensitivity, and that rejection sensitivity uniquely predicted unassertive interpersonal behavior beyond parental rejection and internalized homophobia [35]. Expectations of rejection are rooted in perceived stigma, understood as awareness that others engage in heterosexist acts or endorse heterosexist beliefs [37]. Increased perceived stigma corresponds to greater vigilance during exchanges with dominant social groups [11,14]. Meyer [11] situated vigilance within minority stress as a recurring demand in the everyday lives of sexual minority people. Although vigilance can be adaptive by reducing exposure to prejudice, severe discriminatory events and the chronic expectation of rejection may escalate this response into hypervigilance with trauma-like features [38]. Consistent with these observations, qualitative evidence shows that hypervigilance is common across home, work, and community settings, is accompanied by anxiety, fear, and exhaustion, and often entails self-monitoring and social withdrawal [39]. In a longitudinal, population-based cohort of Swedish sexual minority young adults, hypervigilance measured with the LGBTQ-Hypervigilance Scale [38] mediated the prospective link between perceived discrimination and later internalizing symptoms, explaining about 40% of the effect on anxiety and 27% on depression [34]. Moreover, hypervigilance has

been proposed to play a role in development and maintenance of obsessive-compulsive disorders (OCD) for which sexual minority populations seem to be at greater risk [40].

**Resilience factors.** Meyer later anchored resilience explicitly within the minority stress model, arguing that coping and social support can buffer stressor effects and thereby prevent or attenuate adverse health outcomes [41]. He distinguished resilience at two levels. At the individual level, resilience refers to personal attributes and skills that enable effective coping. In LGBQ research this has been operationalized as identity *pride*, reflected in earlier work on identity affirmation in the Lesbian, Gay, and Bisexual Identity Scale (LGBIS) [42] and introduced as a resilience element within the minority stress model through Testa et al.'s development of the Gender Minority Stress and Resilience measure [13].

Community resilience, by contrast, denotes the capacity of communities to equip individuals with resources for managing minority stress [41]. In practice this includes shared norms and values, visible role models, and structured opportunities for social support, alongside tangible assets such as community centers, affirming health services and support groups, hotlines, accessible information, and protective legal and policy environments arising from community advocacy (for example marriage equality and anti-bullying policies) [41]. Intangible resources include reframing dominant norms through minority perspectives, such as revising life goals and markers of success [41]. Community resilience depends on the social identification as a sexual minority and to *community connectedness* [41]. To measure how much access individuals have to community resilience, these ideas are reflected through different operationalizations, for example the identity centrality subscale of the LGBIS and the Connectedness to the LGBT Community Scale by Frost and Meyer [43].

## Empirical support and extensions of the model

Empirical research supports the minority stress model. Studies have linked perceived discrimination to various mental disorders in the United States [44], and this association is particularly pronounced in individuals identifying as homosexual or bisexual [45]. Thus, minority stress constructs have been correlated with social anxiety and other mental health issues in queer populations [46,47]. Proximal stressors are linked to poorer psychosocial adjustment and lower quality of life among sexual minority people with chronic illness or disability [48]. Moreover, they moderate the association between workplace harassment and somatic symptoms [49]. Recent evidence further highlights that structural stigma, defined as societal level conditions and policies, consistently contributes to poorer mental, behavioral, and physical health outcomes among LGBTQ+ individuals [50].

However, several adaptations and extensions of the minority stress model have been proposed [51]. Among the most popular extensions is the psychological mediation framework by Hatzenbuehler which specifies the cognitive, affective, and interpersonal processes through which stigma-related stress translates into psychopathology [52]. This framework highlights pathways such as emotion dysregulation, rumination, rejection sensitivity, social isolation, and hopelessness as mediators between stigma exposure and mental health [52].

A further extension builds on Meyer's minority stress model [11] and Hatzenbuehler's psychological mediation framework [52] by incorporating the rejection sensitivity model [36]. While expectations of rejection are part of Meyer's minority stress model, this framework argues the complexity of rejection-related processes and clarifies the underlying mechanism through which proximal stress leads to symptoms, including biases in attention and along with varied anticipatory affect [36].

It is also important to note complementary theory to Meyer's minority stress model [11]. Herek's sexual stigma framework conceptualizes stigma as a societal phenomenon expressed in institutions and individual attitudes [15]. It distinguishes structural stigma (heterosexism), enacted stigma (ostracism, discrimination and violence), felt stigma (anticipated devaluation), and internalized stigma (acceptance of stigma into the self-concept). The framework applies to both sexual minorities and heterosexuals, linking sexual prejudice in majority groups with self-stigma in minority groups [15].

For this review, we use Meyer's minority stress model [11,12,41] as the organizing framework because of its wide uptake and empirical support, while being inclusive in regard to constructs considered to reflect these dynamic developments in the field.

## Need for scoping review

The growing recognition of sexual identity minority stress has resulted in the development of numerous measurement instruments. However, this proliferation presents challenges in selecting appropriate tools due to the multitude of options, each with its unique strengths and limitations. Currently, a comprehensive synthesis of the available instruments is lacking. While a narrative review by Morrison et al. [53] provided insights into the psychometric properties of measures assessing discrimination against sexual minorities, it lacks the systematic methodological rigor of a scoping review. Their work was focused on the negative aspects of minority stress, specifically discrimination and stigmatization.

The aim of this scoping review is to present a comprehensive overview of the various instruments used to assess minority stress in LGBQ individuals. By systematically mapping and analyzing existing instruments that measure different aspects of minority stress, this review contributes significantly to a more harmonized approach for assessing minority stress factors in this population. This will contribute to Peters and Crutzen's call for psychometrics to provide transparent construct definitions in order to counteract the measurement crisis stemming from jingle-jangle fallacies [54]. The jingle-jangle fallacy is the discrepancy between construct and measurement instrument, which occurs when two measurement instruments that are supposed to assess the same construct actually measure different attributes ('jingle'), while two instruments that were developed for different constructs actually measure the same attribute ('jangle') [54,55]. Additionally, our scoping review aims to identify gaps in the current research on minority stress measurement tools for LGBQ individuals.

This scoping review serves as a valuable resource for clinicians and researchers when selecting appropriate tools to assess minority stress. Scoping reviews are increasingly recognized not only for mapping existing research but also for their usefulness in setting research priorities and reducing research waste [56,57]. By providing a detailed overview of the available instruments and their characteristics, it facilitates informed decision-making when choosing the most suitable tool for a specific research or clinical context. Eventually, this review might be used to improve interventions and policies designed to promote the well-being of LGBQ populations by accurately measuring minority stress [1,58,59]. In this scoping review, we focus on sexual-identity minority stress among LGBQ adults; gender-identity-related stressors [13,51] are outside the scope.

## Methods

This review was conducted and reported in accordance with the Preferred Reporting Items for Systematic reviews and Meta-Analyses extension for Scoping Reviews (PRISMA-ScR) guidelines [60]. The PRISMA-checklist for scoping reviews is provided in Supplement S1 File. A corresponding a-priori-protocol was registered on PROSPERO (ID: CRD42021257995).

### Data sources

An experienced information scientist (JS) developed a literature search string (Supplement S2 File) which was peer-reviewed by another information specialist. The search was conducted across Medline, Embase, PsycInfo, PSYNDEX-plus, PTSDpub and Web of Science. Additionally, a hand search for grey literature in the GESIS Open Access Repository for Measurement Instruments and the GLBTQ+ Alliance in Social and Personality (GASP) Measures Database was performed and the reference lists of included articles were examined for further relevant publications. The search started in August 2021 and was last updated in February 2025.

### Study selection

A two-staged screening process was implemented. The first stage screening was based on titles and abstracts; the second stage was based on full texts. Every reference was screened by two independent reviewers. Every inclusion in

the title abstract screening by one screener was included for the full text screening. At the second stage, conflicts on the in- and exclusion of references were resolved in consensus by thorough discussion between the involved screeners and, if no consensus could be reached, through discussion with all screeners (MMK, SC, SF, AJ, MSix), with resolution by majority vote. The selection process was recorded in sufficient detail to complete a PRISMA flow diagram (Fig 2).

The following inclusion criteria were employed:

• Population: Adult homosexual, bisexual and queer persons

• Constructs measured with the instruments: Discrimination, Rejection, Victimization, Stigmatization, Internalized Homophobia, Negative Future Expectations, Concealment, Social Support, and Pride

• Type of instrument: Self-report questionnaires

• Types of studies: Psychometric studies.

Only articles in English and German were considered. No restrictions were applied with regard to the year of publication or publication medium.

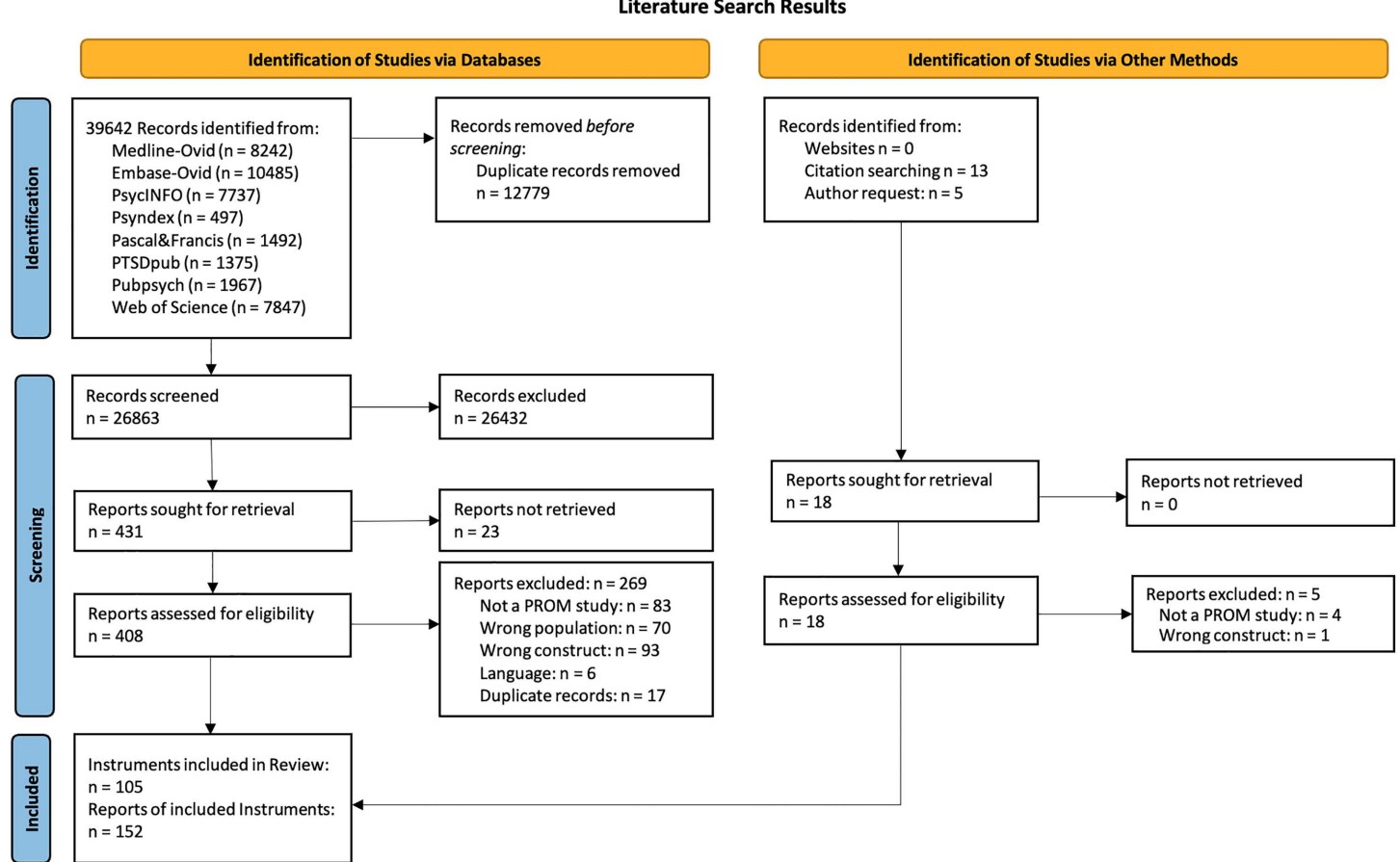

**Fig 2. PRISMA 2020 flow diagram for systematic searches.**

## Data extraction

We used a data extraction form for instrument characteristics, which was piloted on 10 studies in the review. This included the instruments' languages, constructs and subscales, target populations, and number of items. One study author (MMK) extracted the characteristics data which was checked for completeness and correctness by another author. The data extraction sheet is provided in Supplement S3 File.

## Data analysis

Instruments were categorized in accordance with the main constructs of the minority stress model they measured, as specified by their psychometric studies. If an instrument claimed to assess two main constructs, e.g., stigmatization and concealment, this instrument was included in both categories, resulting in non-mutually exclusive categories. The terms 'homophobia', 'homonegativity' and 'heterosexism' are used interchangeably in this analysis, reflecting a notable shift from the use of 'homophobia' to 'homonegativity' and 'heterosexism', signifying a broader conceptualization of negative attitudes towards non-heterosexuality [28,29].

An Upset Plot was generated in R (version 4.5.1) using the UpSetR package (version 1.4.0) [61,62] to visualize construct overlap across instruments. Additionally, a heatmap was generated using ggplot2 (version 3.5.1) [63] to visualize the temporal distribution of measurement instruments across constructs. Constructs were coded dichotomously according to Meyer's minority stress model [11], based on the construct and subscale labels reported in the psychometric studies.

Furthermore, instruments were categorized based on their validation populations. For analysis purposes only, studies on men who have sex with men (MSM) or women who have sex with women (WSW) were categorized as psychometric studies on homosexual and bisexual persons. Instruments subjected to multiple population validations were classified according to the most comprehensive psychometric study available.

Language validations of measurement instruments, defined as translation and cross-cultural adaptation of an existing instrument followed by psychometric testing, were not treated as separate entities in this analysis to prevent an overrepresentation of the same instrument. This decision was made considering that there could be many language validations of a single questionnaire, which, if counted separately, could artificially inflate the number of categories. Contrastingly, if short versions of a questionnaire existed, they were considered separately because they often measure slightly different constructs or dimensions of the same construct, reflecting the necessary adjustments in item selection and emphasis.

## Results

Initially, we identified a total of 39,642 records from all databases, of which 105 instruments with 152 references fulfilled all inclusion criteria (Fig 2). The selected instruments are presented in Table 1.

Most psychometric studies were reported in peer-reviewed journals. However, 16 instruments were documented in grey literature. Seven were available only as theses:

1. Emerging Adult Inventory of Minority Stress (EAIMS) [73]

2. The Positive Aspects of Nonheterosexuality Questionnaire (PANQ) [96]

3. The Internalized Heterosexist Racism Measure (IHRM) [115]

4. The Gay Identity-based Shame and Pride Scales [149]

5. Internalized Sexual Prejudice Scale (ISPS) [158]

6. Chinese and Chinese American Sexual Minority Women Internalized Heterosexism (CCSMW-IH) Scale [173]

7. Bisexual Microaggressions Scale [200].

**Table 1. Measurement Instruments on Minority Stress and its Components.**

| No. | Questionnaire (Abbreviation) | Language [a] | Construct (Subscale) | Validation Population [b] | No. of Items [c] (Subscale) | Study Context | Psychometric Study |
|---|---|---|---|---|---|---|---|
| Instruments targeting at least lesbian, gay and bisexual people (LGB+) [d] | | | | | | | |
| 1 | The Daily Hetero-sexist Experiences Questionnaire (DHEQ) | English | Minority Stress (Gender Expression, Vigilance, Parenting, Harassment and Discrimination, Vicarious Trauma, Family of Origin, HIV/AIDS, Victimization, Isolation) | LGBQT | 50 (6 + 6 + 6 + 6 + 6 + 6 + 5 + 5 + 4) | Study 1: Washington State, USA Study 2–3: USA, online | [64] |
| | | Polish | | | | Poland, online | [65] |
| | | Spanish | | | | Spain, online | [66] |
| 2 | Sexual Minority Stress Scale (SMSS) [e] | Polish | Minority Stress (Internalized Homophobia, Expectations of Rejection, Concealment, Sexual Minority Negative Events, Satisfaction with Outness) | LGB | 58 (10 + 6 + 6 + 26 + 10) | Poland, online | [67] |
| 3 | LGBT Minority Stress Measure | English | Minority Stress (Identity Concealment, Everyday Discrimination/ Microaggressions, Rejection Anticipation, Discrimination Events, Internalized Stigma, Victimization Events, Community Connectedness) | LGBT | 25 (4 + 4 + 4 + 4 + 3 + 3 + 3) | USA, online | [68] |
| | | | Minority Stress (Identity Concealment, Rejection Anticipation, Internalized Stigma, Victimization events, Community Connectedness) | MSM | 19 (4 + 4 + 3 + 3 + 5) | Study 1: Delta/ Lagos, Nigeria, in-person interviews Study 2: Abuja, Delta, Lagos or Plateau, Nigeria, in-person, interviewer-assisted survey | [69] |
| 4 | Minority Stress Scale (MSS) | Italian | Minority Stress (Structural Stigma, Enacted Stigma, Expectations of Discrimination, Expectations of Discrimination From Family Members, Sexual Orientation Concealment, Internalized Homophobia Toward Oneself, Internalized Homophobia Toward Others, Stigma Awareness) | Gay or Bisexual Men | 50 [f] (3 + 3 + 12 + 3 + 7 + 6 + 6 + 3) | Italy, online | [70] |
| | | Croatian | | Cisgender LGB | 32 (3 + 3 + 8 + 3 + 3 + 4 + 5 + 3) | Croatian, online | [71] |
| 5 | Military Minority Stress Scale (MMSS) | English | Military Minority Stress | LGBT | 13 | Active-duty military personnel in USA, online | [72] |
| 6 | Emerging Adult Inventory of Minority Stress (EAIMS) | English | Sexual and Gender Minority Stress (Distal Stress, Internalized Sexual and Gender Minority Negativity, Relational Vigilance, Identity Instability, Sexual and Gender Minority-Related Apprehension) | LGB | 19 (6 + 4 + 4 + 2 + 3) | College students in USA, online | [73] |

*(Continued)*

| No. | Questionnaire (Abbreviation) | Language [a] | Construct (Subscale) | Validation Population [b] | No. of Items [c] (Subscale) | Study Context | Psychometric Study |
|---|---|---|---|---|---|---|---|
| 7 | Lesbian, Gay, and Bisexual Identity Scale (LGBIS) [g] | English | Concealment and Negative Identity (Acceptance Concerns, Concealment Motivation, Identity Uncertainty, Internalized Homonegativity, Difficult Process, Identity Superiority, Identity Affirmation, Identity Centrality) | LGBQT | 27 (3 + 3 + 4 + 3 + 3 + 3 + 3 + 5) | University students in USA, online | [42] |
| | | | | | | BDSM practitioners in USA, online | [74] |
| | | | | | | Military veterans in USA, in-person, self-administered survey | [75] |
| | Lesbian, Gay, and Bisexual Identity Scale-German (LGBIS-DE) | German | | Lesbian or Gay | | USA, Germany, online | [76] |
| | | | | MSM and WSW | | Germany, online | [77] |
| | Lesbian, Gay, and Bisexual Identity Scale-Turkish (LGBIS-TR) | Turkish | | LGBQ | | Turkey, online | [78] |
| | Lesbian, Gay, and Bisexual Identity Scale (LGBIS) | Portuguese | Concealment and Negative Identity (Identity Dissatisfaction, Identity Uncertainty, Concealment Motivation, Difficult Process, Identity Centrality, Stigma Sensitivity, Identity Superiority) | LGB | 28 [h] (6 + 4 + 4 + 4 + 4 + 3 + 3) | Portugal, online | [79] |
| | Lesbian, Gay, and Bisexual Identity Scale (LGBIS-CZ-6) | Czech | Concealment and Negative Identity (Acceptance Concerns, Concealment Motivation, Internalized Homonegativity, Difficult Process, Identity Affirmation, Identity Centrality) | LGB+ | 20 (3 + 3 + 3 + 3 + 3 + 5) | Czechia, online | [80] |
| 8 | Sexual Orientation Reflection and Rumination Scale (SRRS) | English | Reflective and Ruminative Thought in the Context of Sexual Minority Identity (Reflection, Rumination, Preoccupation with Others' Perceptions, and Perseveration) | LGBQ | 12 (3 + 3 + 3 + 3) | USA, online | [81] |
| 9 | Measure of Internalized Sexual Stigma for Lesbians and Gay Men (MISS-LG) | Italian | Internalized Sexual Stigma (Identity, Social Discomfort, and Sexuality) | Lesbian or Gay | 17 (5 + 7 + 5) | Rome, Italy and online | [82] |
| | | Traditional Chinese | | LGB | | Kaohsiung, Taiwan, in-person, self-administered survey | [83] |
| 10 | Heterosexist Harassment, Rejection, and Discrimination Scale (HHRDS) | English | Heterosexist Harassment, Rejection, and Discrimination (Harassment and rejection, Workplace and School Discrimination, Other Discrimination) [i] | Lesbian | 14 (7 + 4 + 3) | LGB Pride Festival in the Midwest, USA, self-administered paper survey | [84] |
| | | | | LGBTQ People of Color | | Racial/ ethnic minorities in the USA, online | [85] |
| | | | | Cis and Trans LGB+ (18–29 years) | | USA, online | [86] |
| | | | Heterosexist Harassment, Rejection, and Discrimination | LGB+ | 11 | USA, online | [87] |

*(Continued)*

| No. | Questionnaire (Abbreviation) | Language [a] | Construct (Subscale) | Validation Population [b] | No. of Items [c] (Subscale) | Study Context | Psychometric Study |
|---|---|---|---|---|---|---|---|
| 11 | Perceived Online Heterosexism Scale (POHS) | English | Experiences of Online Heterosexism (Heterosexist Cyberaggression, Online Heterosexist Stereotyping, Online Exposure to Systemic Heterosexism, Heterosexist Online Media) | LGBTQ+ | 22 [j] (4 + 4 + 8 + 4) | USA, online | [88] |
| 12 | Self-Acceptance of Sexuality Inventory (SASI) | English | Self-Acceptance (Self-Acceptance of Sexuality, Difficulties with Self-Acceptance of Sexuality) | LGBQ+ | 10 (5 + 5) | UK, North America, Europe, Oceania, South/ Central America, Asia, Africa, other countries of origin, online | [89] |
| 13 | Factors of Self-Acceptance – Sexual and Gender Identities (FSA-SGI) Scale | English | Self-Acceptance, Safety, Connection | LGBTQ+ | 18 (7 + 5 + 6) | USA, online | [90] |
| | | | | LGBQ+ | | USA, online | [91] |
| 14 | Everyday Discrimination Scale for Sexual Minority – Portuguese (EDS-PT-SM) | Portuguese | Discrimination (Unfair Treatment, Personal Rejection) | LGBQ | 8 (4 + 4) | Portugal, online | [92] |
| | Everyday Discrimination Scale (EDS) | English | Discrimination | LGBTQ+ | 9 | USA, online | [93] |
| | | | | LGBTQ | | USA, online | [94] |
| 15 | Discrimination-Related Vigilance Scale (DRVS) | English | Vigilance (Preparation, Caution) | | 6 (3 + 3) | | |
| 16 | LGBTQ-Hypervigilance Scale | English | Hypervigilance, Occurrence (where and around whom) | LGBTQ | 13 + 12 | USA, online | [38] |
| 17 | Coming Out Vigilance (COV) Measure | English | Coming Out Vigilance | Sexual and Gender Minorities | 3 | Current or previous affiliation with the Church of Jesus Christ of Latter-day Saints, USA, online | [95] |
| 18 | Positive Coming Out Responses (PCOR) Measure | English | Positive Coming Out Responses | | 8 | | |
| 19 | The Positive Aspects of Nonheterosexuality Questionnaire (PANQ) | English | Positive Aspects | LGB | 5 | Current or previous affiliation with the Church of Jesus Christ of Latter-day Saints, USA, online | [96] |
| 20 | A Multifactor Lesbian, Gay, and Bisexual Positive Identity Measure (LGB-PIM) | English | Positive Lesbian, Gay and Bisexual Identity (Self-Awareness, Authenticity, Community, Intimacy, Social Justice) | Male or Female LGB | 25 (5 + 5 + 5 + 5 + 5) | USA, online | [97] |
| | | German | | LGB | | Austria, Germany, Switzerland, online | [98] |
| | | Turkish | | LGBTQ+ | | Turkey, online | [99] |
| 21 | Sexual Orientation Concealment Scale (SOCS) | English | Concealment | LGB | 6 | University students in the USA, online | [100] |

*(Continued)*

| No. | Questionnaire (Abbreviation) | Language [a] | Construct (Subscale) | Validation Population [b] | No. of Items [c] (Subscale) | Study Context | Psychometric Study |
|---|---|---|---|---|---|---|---|
| 22 | The Lesbian, Gay, Bisexual-Visibility Management Scale (LGB-VMS) | English | Visibility Management (Active Behavioral, Inhibitive Behavioral, and Setting) | LGB | 28 (13 + 11 + 4) | USA, online | [101] |
| 23 | Nebraska Outness Scale (NOS) | English | Outness (Concealment, Disclosure) | LGB | 10 (5 + 5) | USA, online | [102] |
| | | | | | | USA, online | [93] |
| 24 | Fear of Heterosexism Scale (FoHS) | English | Fear of Heterosexism | LGBQ | 16 | Tasmania, no information | [103] |
| 25 | Revised Internalized Homophobia (IHP-R) Scale | English | Internalized Sexual Stigma | LGB | 5 | California, USA, self-administered paper survey | [15] [k] |
| | | Thai | | | | Thailand, online and in-person surveys | [104] |
| 26 | Self-Stigma Scale–Short Form (SSS–S) | Chinese | Self-Stigma (Affective, Behavioral, Cognitive) | LGB | 9 (3 + 3 + 3) | Hong Kong, online | [105] |
| 27 | Stigma-Consciousness Questionnaire (SCQ) for Gay Men and Lesbians | English | Stigma Consciousness (Stigma Experiences, Stigma Beliefs) | Lesbian or Gay | 10 (No information) [l] | Studies 1–2, 4: Psychology students in USA, self-administered paper survey Studies 3, 5: Gay Pride Festival in California, USA, self-administered paper survey | [106] |
| | | English, Spanish | | | 8 (No information) | USA, Spain, online | [107] |
| | Stigma Consciousness Questionnaire – European Portuguese Version (SCQ-PT) | Portuguese | | LGBQ | 10 (6 + 4) | Portugal, online | [108] |
| 28 | Perceived Familial Stigma of Sexuality Questionnaire (PFSSQ) | English | Perceived Familial Stigma of LGBQ Sexuality | LGBQ | 30 | USA, online | [109] |
| 29 | Sexual Orientation Experiences of Discrimination-9 (SOEOD-9) | English [m] | Experiences of Discrimination (Proximal Discrimination, Distal Discrimination) | LGBTQI [n] | 9 (4 + 5) | Santa Marta, Colombia, self-administered paper survey | [110] |
| 30 | Sexual Orientation Experiences of Discrimination-5 (SOEOD-5) | | Experiences of discrimination | | 5 | | |
| 31 | The Wright Internalized Homophobia Scale | English | Internalized Homophobia | LGB | 9 | USA, online | [111] |

*(Continued)*

| No. | Questionnaire (Abbreviation) | Language [a] | Construct (Subscale) | Validation Population [b] | No. of Items [c] (Subscale) | Study Context | Psychometric Study |
|---|---|---|---|---|---|---|---|
| 32 | Internalized Homophobia Scale (IHS) [o] | English | Internalized Homophobia | LGB | 9 | Street fair in California, USA, in-person interviews | [112] |
| | | | | | | USA, online | [93] |
| | | Chinese | | | | Southwest China, online | [113] |
| | | Turkish | | | 10 | Ankara/ Istanbul, Turkey, no information | [114] |
| 33 | The Internalized Heterosexist Racism Measure (IHRM) | English | Internalized Heterosexist Racism (Negative Messages, Intersectional Minority Stress and Reactivity, Assimilation of Beauty and Self-Expression Standards, Internalized Inferiority, Internalized Isolation and Ostracism, Intersectional Invisibility) | LGBTQ+ People of Color | 48 (No information) | USA, online | [115] |
| 34 | Sexual Minority-External and Internal Shame Scale (SM-EISS) | Portuguese | Shame related to Sexual Orientation (External Shame, Internal Shame) | LGB+ | 8 (4 + 4) | Portugal, online | [116] |
| 35 | The Homophobic Bullying Scale (HBS) | English | Homophobic Bullying | LGB | 9 | Workers employed full-time in USA, online | [117] |
| 36 | The Lesbian, Gay, Bisexual and Transgendered Climate Inventory (LGBTCI) | English | LGBT Workplace Climate | LGBT Workers | 20 | USA, self-administered paper survey | [118] |
| | | Spanish | Workplace Climate of Support and/or Hostility (Support and Non-Hostility) | | 15 (10 + 5) | Spain, online and self-administered paper survey | [119] |
| 37 | Brief Sense of Community Scale (BSCS) | English | Sense of Community | LGBT | 8 | Georgia/ South Carolina, USA, online | [120] |
| 38 | Psychological Sense of LGBT Community Scale (PSOC-LGBT) | English, Spanish | Sense of Community (Influence, Shared Emotion, Needs Fulfillment, Membership, Communities Existence) | LGBQ | 22 | West Coast, USA, online | [121] |
| 39 | Connectedness to the LGBT Community Scale | English | Community Connectedness | LGBT and Heterosexual | 8 | New York City, USA, in-person, interviewer-administered | [43] |
| | | | | Black Sexual Minority Men | 9 | Men living with HIV in Mid-Atlantic, USA, online | [122] |
| 40 | LGBTQ Belongingness Attainment Scale (LGBTQ BAS) | English | Community Connectedness (Connectedness, Affiliation, Companionship) | LGBTQ | 18 (6 + 6 + 6) | Washington, DC, USA, in-person, interviewer-administered | [123] |
| 41 | Relational Health Indices–Community (RHI-C) | English | LGBTQ+ Community Connectedness | LGBTQ+ People of Color | 10 | USA, online | [124] |

*(Continued)*

**Table 1.** (Continued)

| No. | Questionnaire (Abbreviation) | Language [a] | Construct (Subscale) | Validation Population [b] | No. of Items [c] (Subscale) | Study Context | Psychometric Study |
|---|---|---|---|---|---|---|---|
| 42 | Multidimensional Scale of Perceived Social Support (MSPSS) | English | Perceived Social Support (Family, Friends, Significant Others) | LGBTQ People of Color | 12 (4 + 4 + 4) | USA, online | [125] |
| | | | | Sexual Minority Women (aged 55 and older with disabilities) | | USA, online | [126] |
| 43 | LGBTQ+ Community Resilience and Inequity Scale (LGBTQ+ CRIS) | English | Community Resilience and Inequity (Community Resilience, Community Inequity) | LGBTQ+ | 20 (10 + 10) | College students in USA, online | [127] |
| 44 | Courage to Challenge Scale | English | Resilience | LGBT | 18 | South Florida, USA, online and self-administered paper survey | [128] |
| 45 | Gay Community Stress Scale (GCSS) | English | Gay Community Stress (Sex, Status, Competition, Exclusion) | Gay and Bisexual Men | 20 (6 + 4 + 7 + 3) | USA, Sweden, online | [129] |
| | | | | | 3 versions: 20 (6 + 4 + 7 + 3) 8 (2 + 2 + 2 + 2) 4 (1 + 1 + 1 + 1) | USA, Sweden, online | [130] |
| | | Dutch | | LGB | Male version: 16 (6 + 3 + 5 + 2) Female version: 12 (4 + 2 + 4 + 2) | Netherlands, online | [131] |
| | | Chinese | Gay Community Stress (Sex, Status, Competition, Exclusion, and Externals) | Gay and Bisexual Men | 32 (6 + 5 + 6 + 8 + 7) | Kaohsiung, Taiwan, self-administered paper survey | [132] |
| 46 | Gay-Related Rejection Sensitivity (RS) Scale [p] | English | Rejection Sensitivity | Gay | 14 | New York City, USA, self-administered paper survey | [35] |
| | | | | Gay and Bisexual | 14 | USA, Sweden, no information | [133] |
| | | | | LGBTQ | 12 | USA, online | [93] |
| Instruments targeting Homosexual Men and Women | | | | | | | |
| 47 | Lesbian and Gay Identity Scale (LGIS) | English | Need for Privacy, Need for Acceptance, Internalized Homonegativity, Difficult Process, Identity Confusion, Superiority | Lesbian or Gay | 6 + 5 + 5 + 5 + 4 + 2 | Same-sex couples in East Coast, USA, self-administered paper survey | [134] |
| 48 | The Multifactor Internalized Homophobia Inventory (MIHI) | English, Italian | Internalized Homophobia (Fear of Coming Out, Regret about Being Homosexual, Moral Condemnation, Gay-Lesbian Parenting, Integration in the Homosexual Community, Counter-Prejudicial Attitudes, Homosexual Marriage, Stereotypes) | Lesbian or Gay | 77 [q] (21 + 11 + 15 + 4 + 7 + 7 + 2 + 7) | Italy, self-administered paper survey | [135] |

*(Continued)*

 

| No. | Questionnaire (Abbreviation) | Language [a] | Construct (Subscale) | Validation Population [b] | No. of Items [c] (Subscale) | Study Context | Psychometric Study |
|---|---|---|---|---|---|---|---|
| 49 | The Short Internalized Homonegativity Scale (SIHS) [r] | English | Internalized Homonegativity (Public Identification as Homosexual, Sexual Comfort with Homosexual People, Social Comfort with Homosexual People) | Gay | 12 (4 + 4 + 4) | 40 countries, online | [136] |
| | | Spanish | | Lesbian or Gay | 13 (5 + 4 + 4) | Spain, online | [137] |
| 50 | Internalized Homonegativity Inventory (IHNI) | English | Internalized Heterosexism (Personal Homonegativity, Gay Affirmation, Morality of Homosexuality) | MSM | 23 | Midwest, USA, self-administered paper survey | [28] |
| | | | | | | African-American MSM in Alabama, Georgia, Louisiana, Mississippi, North Carolina, or South Carolina, USA, self-administered paper survey | [138] |
| | | | | | | USA, online | [139] |
| | | Russian | Personal Homonegativity (Homonegativity, Acceptance of one's own Homosexuality) | Lesbian or Gay | 10 (5 + 5) | Cities in Russia, online | [140] |
| 51 | Gay and Lesbian Oppressive Situations Inventory – Frequency (GALOSI-F) | English | Couples Issues, Danger to Safety, Exclusion, Rejection and Separation, Internalized Homonegativity, Restricted Opportunities and Rights, Stigmatizing and Stereotyping, Verbal Harassment and Intimidation | Lesbian or Gay | 49 (4 + 5 + 9 + 10 + 3 + 11 + 7) | USA, online | [141] |
| 52 | Gay and Lesbian Oppressive Situations Inventory – Effect (GALOSI-E) | | | | 47 (3 + 6 + 10 + 9 + 3 + 8 + 8) | | |
| Instruments targeting Men who have Sex with Men (MSM) and/or Homosexual and Bisexual Men | | | | | | | |
| 53 | Internalized Homophobia Scale (IH Scale) [s] | English | Internalized Homophobia (Public Identification as Gay, Perception of Stigma Associated with Being Gay, Social Comfort with Gay Men, Moral and Religious Acceptability of Being Gay) | MSM | 26 (10 + 6 + 6 + 4) | Midwestern city in USA, self-administered paper survey | [31] |
| | | English, Luganda | Internalized Homophobia (Personal Comfort with Being Gay, Social Comfort with Other Gay People, Public Identification as Being Gay) | | 8 | Kampala, Uganda, In-person, interviewer-administered | [142] |
| 54 | Reactions to Homosexuality Scale (RHS) [t u] | English, Spanish | Internalized Homonegativity (Personal Comfort with a Gay Identity, Social Comfort with Gay Men, Public Identification as Gay) | MSM | 7 (3 + 2 + 2) | USA, online | [143] |
| | | 25 languages | | | | 38 European countries, online | [144] |
| | | Brazilian Portuguese | | | | Brazil, online | [145] |

*(Continued)*

| No. | Questionnaire (Abbreviation) | Language [a] | Construct (Subscale) | Validation Population [b] | No. of Items [c] (Subscale) | Study Context | Psychometric Study |
|---|---|---|---|---|---|---|---|
| 55 | Homosexuality-Related Stigma Scale | Vietnamese | Stigmatization (Enacted Homosexual Stigma, Perceived Homosexual Stigma, Internalized Homosexual Stigma) | MSM | 28 (9 + 11 + 8) | Hanoi, Vietnam, In-person, interviewer-administered | [146] |
| 56 | Subjective Scale of Stigma and Discrimination (SISD) | Spanish | Stigmatization and Discrimination (Stigma and Discrimination Experiences, Disadvantages in the Presence of Authorities, Discrimination at Work, Expression of Sexual Identity, Institutional Exclusion and Rights Denial, Religious Discrimination) | Gay | 23 (5 + 3 + 6 + 3 + 3 + 3) | Study 1: Antofagasta, Chile, self-administered paper survey Study 2: Arica/ Valparaíso/ Santiago, Chile, no information | [147] |
| 57 | Multidimensional Sexual Identity Stigma (MSIS) Scale | English, Xhosa | Stigmatization and Concealment (Enacted, Orientation Concealment, Anticipated, and Internalized Stigma) | MSM | 23 (4 + 5 + 9 + 5) | Cape Town/ Port Elizabeth, South Africa, no information | [148] |
| 58 | The Gay Identity-based Shame and Pride Scales | English | Gay Identity-based Shame, Gay Identity-based Pride | Gay | 25 + 25 | USA, self-administered paper survey | [149] |
| 59 | Internalized Homophobia Measure | English | Internalized Homophobia (Desire to be Heterosexual, Fears of Coming out, Worries about Stereotypes) | Young MSM | 8 | Chicago, USA, online | [37] |
| 60 | Perceived Stigma Measure | | Stigmatization | | 7 | | |
| 61 | Nungesser Homosexual Attitudes Inventory (NHAI) | English | Internalized homophobia (Personal Homonegativity, Global Homonegativity, Disclosure) [v] | Gay | 34 (10 + 12 + 12) [w] | West Coast, USA, self-administered paper survey | [32] |
| | | | | | | USA, online | [150] |
| 62 | Shidlo-revised Nungesser Homosexual Attitudes Inventory (NHAI-SR) | | | | 36 (14 + 9 + 13) | Study 1: Northeastern, USA, self-administered paper survey | [29] |
| | | | | | | Study 2: New York City, USA, self-administered paper survey + telephone or in-person interview | |
| 63 | Multi-Axial Gay Men's Inventory-Men's Short Version (MAGI-MSV) [x] | English | Internalized Homophobia (Homosexual Self-Assurance and Worth, Public Appearance of Homosexuality, Extreme or Maladaptive Measure to eliminate Homosexuality, Impacts of HIV/AIDS on Homosexuality) | Gay | 14 (7 + 3 + 2 + 2) | Sexual risk-reduction counseling in New York City, USA, in-person, interviewer-administered | [151] |
| | | | | | 15 (No information) | Sexual risk-reduction counseling in New York City, USA, in-person, interviewer-administered | [152] |

*(Continued)*

| No. | Questionnaire (Abbreviation) | Language [a] | Construct (Subscale) | Validation Population [b] | No. of Items [c] (Subscale) | Study Context | Psychometric Study |
|---|---|---|---|---|---|---|---|
| 64 | The China MSM Stigma Scale [y] | Chinese | Homosexuality Stigma (Perceived Stigma, Enacted Stigma) | MSM | 9 (3 + 6) | Shanghai, China, in-person, interviewer-administered | [18] |
| 65 | The Neilands sexual stigma scale [z] | English, Kiswahili | Homosexuality Stigma (Perceived Stigma, Enacted Stigma) | Gay and Bisexual MSM | 8 (3 + 5) | Sexual risk-reduction counseling Kenya and Nairobi, audio computer-assisted self-interview | [153] |
| 66 | Experienced and Anticipated Sexual Stigma Scale in Health-care Services (EASSSiHS) | English [aa] | Experienced and Anticipated Sexual Stigma in Health-Care Services (Experienced Stigma, Anticipated Stigma) | Gay and Bisexual Men | 6 (4 + 2) | Kaohsiung, Taiwan, self-administered paper survey | [154] |
| 67 | The Enacted Sexual Stigma Experiences Scale in Military Service (ESSESiMS) | English | Experience of Enacted Sexual Stigma | Gay and Bisexual Men | 5 | Men with military service experience in Taiwan, self-administered paper survey | [155] |
| 68 | Internalized Homophobia Scale for Gay Chinese Men | Chinese | Internalized Homophobia (Internalized Heteronormativity, Family-Oriented Identity, and Socially-Oriented Identity) | Gay | 11 | China, online and self-administered paper survey | [156] |
| 69 | Gay Male Heterophobia Scale | English | Heterophobia (Unease/Avoidance, Disconnectedness, Expected Rejection) | Gay | 20 | USA, online | [157] |
| 70 | Internalized Sexual Prejudice Scale (ISPS) | English | Internalized Sexual Prejudice (Civil, Gender, Group Identity, Moral, Sexual Identity) | MSM | 24 (7 + 5 + 4 + 4 + 4) | USA, online | [158] |
| 71 | Experiences and Perceptions of Discrimination Scale | English | Discrimination and Attributions (Globality/importance, Blame orientation, Internal/external-circumstances, Controllability | MSM | (5 + 5 + 5 + 5) | USA, online | [159] |
| 72 | Sexual Behavior Stigma Scale [bb] | English, local languages | Stigmatization (Stigma from Family and Friends, Anticipated Health-Care Stigma, General Social Stigma) | MSM | 13 (3 + 2 + 8) [cc] | Cameroon, Senegal, Burkina Faso, Côte d'Ivoire, Nigeria, Togo, Lesotho, Eswatini and United States; In-person, interviewer-administered and online (USA only) | [160] [161] [dd] |
| | | Mexican Spanish | | Cis-MSM | | Mexico, online | [162] |
| 73 | Chinese Homosexuality-Related Stigma Scales | Chinese | Stigmatization (Public Homosexual Stigma, Self Homosexual Stigma, Public HIV Stigma) | MSM (in high sexual risk) | 10 + 8 + 7 | Shenzhen, China; in-person, interviewer-administered | [163] |

*(Continued)*

| No. | Questionnaire (Abbreviation) | Language [a] | Construct (Subscale) | Validation Population [b] | No. of Items [c] (Subscale) | Study Context | Psychometric Study |
|---|---|---|---|---|---|---|---|
| 74 | Concealment Stress during Adolescence Sexual minority-related positive identity development (SM-PID) | English | Minority Stress (Conceal-ment Stress, Positive Identity Development) | MSM (16–24 years) | 8 + 6 | Chicago, USA, audio computer-assisted self-interview | [164] |
| 75 | Network Social Capital Scale | English | Social Capital | Gay and Bisexual MSM | 9 | Pride festival attendees in Milwaukee, USA, self-administered paper survey | [165] |
| 76 | Index of Sojourner Social Support (ISSS) | Spanish | Social Support | Gay MSM (Spanish-Speaking Immigrant Latino) | 11 | North Carolina, USA, in-person, interviewer-administered | [166] |
| 77 | Homophobia Management Scale | English | Homophobia Management | Cisgender Sexual Minority Men in Midlife and Older Adulthood | 6 | Baltimore/ Washington DC/ Chicago/ Pittsburgh/ Los Angeles, USA, self-administered paper/ tablet survey | [167] |
| 78 | Gay-Specific Intraminority Stigma Inventory (G-SISI) | English | Intraminority Stress (Age Stigma, Socioeconomic Stigma, Gay Non-Conformity Stigma, Racial Stigma, Gender Expression Stigma, Body Stigma) | Gay | 20 (3 + 3 + 3 + 4 + 4 + 3) | USA, online | [168] |
| 79 | Internalized Bi/ Homophobia Scale (IBHS) [ee] | English | Internalized Bi-/Homophobia | Bisexual MSMW | 13 | USA, online | [169] |
| 80 | Enacted Stigma Scale | English | Enacted Stigma (Discrimination, Interpersonal, Felt Stigma) | Black MSM | 24 (10 + 8 + 6) | USA, online | [170] |
| 81 | Anticipated Stigma Scale | | Anticipated Stigma (Discrimination, Interpersonal, Felt Stigma) | | 17 (5 + 7 + 5) | | |
| Instruments targeting Women who have Sex with Women (WSW) and/or Homosexual and Bisexual Women | | | | | | | |
| 82 | Daily Sexual Minority Stressors Scale (DSMSS) | English | Minority Stress | Lesbian | 8 | USA, online | [171] |
| | | | | | | Same-sex couples in USA, online | [172] |

*(Continued)*

**Table 1.** (Continued)

| No. | Questionnaire (Abbreviation) | Language [a] | Construct (Subscale) | Validation Population [b] | No. of Items [c] (Subscale) | Study Context | Psychometric Study |
|---|---|---|---|---|---|---|---|
| 83 | Chinese and Chinese American Sexual Minority Women Internalized Heterosexism (CCSMW-IH) Scale | English | Internalized Heterosexism (Negative Attitudes and Feelings About Oneself as a Sexual Minority Individual, Negative Attitudes and Feelings Toward Same-Sex Sexual Orientation in General and Toward Other Sexual Minority Individuals, Disclosure and/or Public Identification as Sexual Minority Women, Being 'Out' in social situation as sexual minority women, Heteronormative Filial Piety Beliefs about Women's Roles Among Sexual Minority Women, Negative attitudes about same-sex relationships attraction and sex, Narrowed Sense of Intersectional Identity Among Sexual Minority Women) | Experts on Chinese and Chinese American sexual minority females, Chinese and Chinese American sexual minority issues, and/ or internalized heterosexism [ff] | 49 (7 + 7 + 7 + 7 + 7 + 7 + 7) | Experts only [ff], USA, online | [173] |
| 84 | Lesbian Internalized Homophobia Scale (LIHS) | English | Internalized Homophobia (Connection with Lesbian Community, Public Identification as a Lesbian, Personal Feelings about being a Lesbian, Moral and Religious Attitudes towards Lesbians, Attitudes towards Other Lesbians) | Lesbian | 52 (13 + 16 + 8 + 7 + 8) | USA, self-administered paper survey | [174] |
| | | Italian | | | | Italy, self-administered paper survey | [175] |
| | Russian version of the Lesbian Internalized Homophobia Scale (R-LIH) | Russian | Internalized Homophobia (Connection with Lesbian Communities, Public Identification as a Lesbian, Public Visibility as a Lesbian, Cultural Awareness of Lesbian Communities) | Lesbian | 24 (9 + 7 + 5 + 3) | Attendees of a lesbian rights seminar in Moscow, Russia, self-administered paper survey | [176] |
| 85 | The Self-Identified Lesbian Internalized Homophobia Scale (SLIHS) | English | Internalized Homophobia (Visibility, Connectedness, Self-Acceptance, Judgment) | Lesbian | 36 (10 + 8 + 7 + 13) | USA, no information | [177] |
| 86 | The Internalized Homophobia Scale for Vietnamese Sexual Minority Women (IHVN-W) | Vietnamese | Internalized Homophobia (Self-Stigma I [not normal], Self-Stigma II [Self-Reproach, Wishing away Same Sex Sexuality], Sexual Prejudice) | WSW | 11 (4 + 4 + 3) | Vietnam, online | [178] |
| 87 | A Lesbian Identity Disclosure Assessment (ALIDA-II) | English | Disclosure within Relationships and in Public | Lesbian | 18 | USA and Canada, online | [179] |
| 88 | Sexual Minority Women -Rejection Sensitivity Scale (SMW-RSS) | English | Rejection Sensitivity | Female LBQ | 16 | USA, online | [180] |
| 89 | Sexual Stigma Scale Adapted for Lesbian, Bisexual and Queer Women | English | Sexual Stigma (Perceived and Enacted Sexual Stigma) | Female LBQ | 5 + 7 | Toronto/ Calgary, Canada, online | [181] |

*(Continued)*

| No. | Questionnaire (Abbreviation) | Language [a] | Construct (Subscale) | Validation Population [b] | No. of Items [c] (Subscale) | Study Context | Psychometric Study |
|---|---|---|---|---|---|---|---|
| Instruments targeting Bisexual People | | | | | | | |
| 90 | Anti-Bisexual Experiences Scale (ABES) [gg] | English | Anti-Bisexual Experiences (Sexual Orientation Instability, Sexual Irresponsibility, Interpersonal Hostility) | Bisexual | 17 (8 + 4 + 5) | Study 1: USA, online Study 2: North America, online | [182] |
| 91 | Brief Anti-Bisexual Experiences Scale (Brief ABES) [gg] | English | Anti-Bisexual Experiences (Sexual Orientation Instability, Sexual Irresponsibility, Interpersonal Hostility) | Bisexual/ Non-Monosexual | 8 (3 + 2 + 3) | USA, online | [183] |
| 92 | Positive Bisexual Identity (PBI) Scale | English | Positive Bisexual Identity | Bisexual | 16 | USA, Canada, online | [184] |
| 93 | Bisexual Identity Inventory (BII) | English | Bisexual Identity (Illegitimacy of Bisexual, Anticipated Binegativity, Internalized Binegativity, and Identity Affirmation) | Bisexual | 24 (8 + 5 + 5 + 6) | USA, online | [185] |
| 94 | Chinese Internalized Binegativity Scale (CIBS) [hh] | Chinese | Internalized Binegativity (Illegitimacy, Monosexism, Aversion, Irresponsibility, Family Shame, Identity Affirmation) | Bisexual | 25 (4 + 4 + 4 + 5 + 3 + 5) | Hong Kong, Mainland China, Taiwan, online | [186] |
| Instruments focusing on Microaggressions | | | | | | | |
| 95 | The LGBT People of Color Microaggressions Scale (LGBT-PCMS) | English | Microaggressions (LGBT Racism, POC Heterosexism, LGBT Relationship Racism) | LGBT people of color | 18 (6 + 6 + 6) | USA, online | [187] |
| | | | | LGBT black and indigenous people and people of color | | College students in USA, online | [188] |
| 96 | The LGBTQ + POC Microaggressions Scale-Brief (LGBTQ + PCMS-B) | English | Microaggressions (LGBT Racism, POC Heterosexism, Racism in Dating and Close Relationships) | LGBTQ + people of color | 12 (4 + 4 + 4) | USA, online | [189] |
| 97 | Sexual Orientation Microaggression Inventory (SOMI) | English | Microaggressions (Anti-Gay Attitudes and Expressions, Denial of Homosexuality, Heterosexualism, Societal Disapproval) | LGB | 19 (6 + 3 + 5 + 5) | Midwest, USA, no information | [190] |
| | | Traditional Chinese | | | | Kaohsiung, Taiwan, self-administered paper survey | [191] |
| 98 | The Sexual Orientation Microaggressions Scale (SOMS) | English | Microaggressions [ii] (Micro-invalidations, Assumption of Pathology, Heterosexist Language, Enforcement of Binary Gender Roles, Environmental Microaggressions) | LGBQ | 24 (7 + 5 + 5 + 3 + 4) | USA, online | [192] |
| | | Swedish | | | | Sweden, online | [193] |
| | Brazilian Portuguese Version of the Sexual Orientation Microaggressions Scale (SOMS-br) | Brazilian Portuguese | | LGB+ | | Brazil, online | [194] |
| 99 | Homonegative Microaggressions Scale (HMS) | English | Microaggressions (Assumed Deviance, Second-Class Citizen, Assumptions of Gay Culture, and Stereotypical Knowledge and Behavior) | LGB | 33 [jj] (9 + 11 + 8 + 5) | USA, online | [195] |
| | | | | | | USA, online | [196] |

*(Continued)*

| No. | Questionnaire (Abbreviation) | Language [a] | Construct (Subscale) | Validation Population [b] | No. of Items [c] (Subscale) | Study Context | Psychometric Study |
|---|---|---|---|---|---|---|---|
| 100 | LGBT Microaggression Experiences at Work Scale (LGBT-MEWS) | English | (Workplace Values, Heteronormative Assumptions, Cisnormative Culture) | LGBT | 27 (12 + 9 + 6) | USA, online | [197] |
| 101 | The LGBQ Microaggressions on Campus Scale | English | Microaggressions (Interpersonal LGBQ Microaggressions, Environmental LGBQ Microaggressions) | LGBQ | 20 (15 + 5) | College students in USA, online | [198] |
| 102 | Thai Sexual Orientation Microaggressions Scale (T-SOMG) | Thai | Microaggressions (Interpersonal Microaggressions, Environmental Microaggressions) | LGBQ+ | 18 (9 + 9) | Thailand, online | [199] |
| 103 | Thai Sexual Orientation Microaffirmations Scale (T-SOMF) | | Microaffirmations (Interpersonal Microaffirmations, Environmental Microaffirmations) | | 13 (8 + 5) | | |
| 104 | Bisexual Microaggressions Scale | English | Microaggressions | Bisexual [kk] | 35 | USA, online | [200] |
| 105 | Bisexual Microaggression and Microaffirmation Scales for Women (BMMS-W) | English | Microaggression (Dismissal, Mistrust, Sexualization, Sexual Exclusion, Denial of Complexity) | Bisexual Women | 38 (6 + 3 + 16 + 10 + 3) | USA and Canada, online | [201] |
| | | | Microaffirmation (Acceptance, Social Support, Recognition of Bisexuality and Biphobia, Emotional Support) | Bisexual Female People of Color | 16 (4 + 3 + 6 + 3) | USA and Canada, online | [202,203][ll] |

[a]Categorized as English if not specified.

[b]As reported by the original psychometric study.

[c]Total number reported only when a total score is formed.

[d]Instruments subjected to multiple population validation studies were classified according to the most comprehensive validation available.

[e]Translation of „The Sexual Minority Stress Scale"by Goldblum P, Waelde L, Skinta M, Dilley J. („Unpublished material, personal information"). The original instrument could not be identified in the systematic search. We received no response to a personal contact request to Goldblum.

[f]The total number of items is stated as 50 items [70]. However, the number of items per subscale add up to 43 items.

[g]Adaptation of the Lesbian and Gay Identity Scale (LGIS) by Mohr & Fassinger [134].

[h]The authors reference a 33-item Lesbian, Gay, and Bisexual Identity Scale (LGBIS) as the basis for the language validation, citing Mohr & Daly [204]. However, this study examines the relationship between sexual minority stress and changes in relationship quality. In this paper, an LGBIS from an "unpublished manuscript" by Mohr and Fassinger (2003) is referenced. As it is unclear, if the scale is similar to the 27-item LGBIS is by Mohr & Kendra [42], the scale is listed separately here.

[i]Smith et al. [85] found a different structure consisting of 2 factors (General Harassment/Rejection and Discrimination from Family).

[j]The total number of items is repeatedly stated as 20 items by Keum et al. [88]. However, the subscales add up to 22 item and 22 items are presented.

[k]The original Internalized Homophobia Scale (IHP) was according to Herek et al. [15] originally derived from the Diagnostic and Statistical Manual of Mental Disorders-3rd edition by John Martin. The original scale was not identified during the systematic search.

[l]Pinel [106] indicates the presence of a single factor in the factor analyses, while the theoretical framework encompasses two distinct subscales: Stigma Beliefs and Stigma Experiences.

[m]Although the study was carried out in Colombia, no translation of the original Experiences of Discrimination (EOD) questionnaire is described. The presented items are in English.

[n]Gallo-Barrera et al. [110] describe the target population as "lesbians, gays, bisexuals, and travesties, transgender, transsexual, intersexual, and queer (LGBTTTIQ)". Primarily observed in Brazil, the identity of travestis also finds relevance in various other Latin American nations [205].

[o]The first identified psychometric study on the IHS is by Herek et al. [112]. Regarding instrument development, the authors describe it as an adaptation of a 9-item measure for self-administration, derived from interview items created by Martin and Dean in 1988. The cited reference refers to an unpublished technical report.

[p]Called the Sexual Orientation-Related Rejection Sensitivity Scale (SORS) by Sullivan et al. [93].

*(Continued)*

**Table 1.** (Continued)

q Flebus et al. [135] performed a factor analysis resulting in a 66-item and a 77-item solution. Due to the insignificant differences between these versions, only the latter is described further. It is inconclusive whether the 77-item questionnaire version represents the final version of the MIHI.

r Adaptation of the Internalized Homophobia (IH) Scale by Ross & Rosser [31].

s Later referred to as "The Reactions to Homosexuality scale" by Smolenski et al. [143].

t Adaptation of the Internalized Homophobia (IH) Scale by Ross & Rosser [31].

u First referred to as the The Short Internalized Homonegativity Scale (SIHS) by Tran et al. [144].

v Subscales in Nungesser [32] labeled Self, Other and Disclosure.

w According to Nungesser [32]. In contrast, the Nungesser Homosexual Attitudes Inventory (NHAI) shown in Shidlo [29] contains 33 (9 + 10 + 14) items.

x According to Theodore et al. [151], Shidlo and Hollander (1996, "unpublished manuscript") originally developed the Multi-Axial Gay Men's Inventory-Men's Short Version (MAGI-MSV) as a modified version of the Shidlo-revised Nungesser Homosexual Attitudes Inventory (NHAI-SR). The original MAGI-MSV is described to contain 20 items distributed across two subscales (excluding "Disclosure") compared to the NHAI-SR.

y Based on a scale by Díaz et al. [206]. No psychometric validity study was identified for the original scale.

z Based on the China MSM Stigma Scale by Neilands et al. [18], referred to as The Neilands Sexual Stigma Scale.

aa Although the study was carried out in Taiwan, the presented items are in English. No translation is described.

bb First time named so by Wiginton et al. [162].

cc Different item loadings onto the 3-factor solutions are reported by Augustinavicius et al. [160] depending on the country. The final assignment of items to a factor is unclear and extracted from Wiginton [162].

dd This is a secondary data analysis using the same sample as Augustinavicius et al. [160].

ee Based on the Internalized Homophobia Scale (IHS) by Herek et al. [112].

ff The measurement instrument is in an early stage of development; psychometric validation in a sample of Chinese and Chinese American sexual minority women has not yet been performed.

gg Each item is answered twice: once to rate experiences within the lesbian/ gay community (ABES-LG) and once to rate experiences with heterosexual persons (ABES-H).

hh Based on the Bisexual Identity Inventory (BII) by Paul et al. [185].

ii Most subscales were not psychometrically supported in the Swedish sample, indicating limited applicability in this setting [193].

jj The original version of the Homonegative Microaggressions Scale (HMS) contained 45 items [195] and was later reduced to 33 items [196].

kk In the psychometric study [200], there is a noticeable deviation from the validation population compared to the target population: 37% of respondents identified as bisexual, 17% as gay/lesbian and 46% as heterosexual. However, the content validity study included only bisexual people.

ll In this study, only the subscale "Microaffirmation" of the Bisexual Microaggression and Microaffirmation Scales for Women (BMMS-W) is investigated. Paul [202] refer to this subscale as the "Bisexual Microaffirmation Scale: For Women (BMSFW)".

Four measurement instruments also had psychometric reports in journals:

1. LGBT Minority Stress Measure [68,69]

2. Nebraska Outness Scale (NOS) [93,102]

3. Multi-Axial Gay Men's Inventory (Men's Short Version, MAGI-MSV) [151,152]

4. Bisexual Microaggression and Microaffirmation Scales for Women (BMMS-W) [201–203].

The Nungesser Homosexual Attitudes Inventory (NHAI) [32] and the Shidlo-revised NHAI (NHAI-SR) [29] were originally published as a book and later further validated in a dissertation [150]. On the other hand, the Self-Identified Lesbian Internalized Homophobia Scale (SLIHS) was originally published as a dissertation and later also published in a book [177,207][]. Finally, two instruments, GALOSI-E and GALOSI-F [141], were identified only in conference proceedings that reported detailed psychometric results.

## Constructs

The identified instruments reflect the multidimensional nature of minority stress. Fig 3 presents an UpSet plot illustrating construct coverage across all instruments. Eight instruments explicitly assess minority stress as a comprehensive

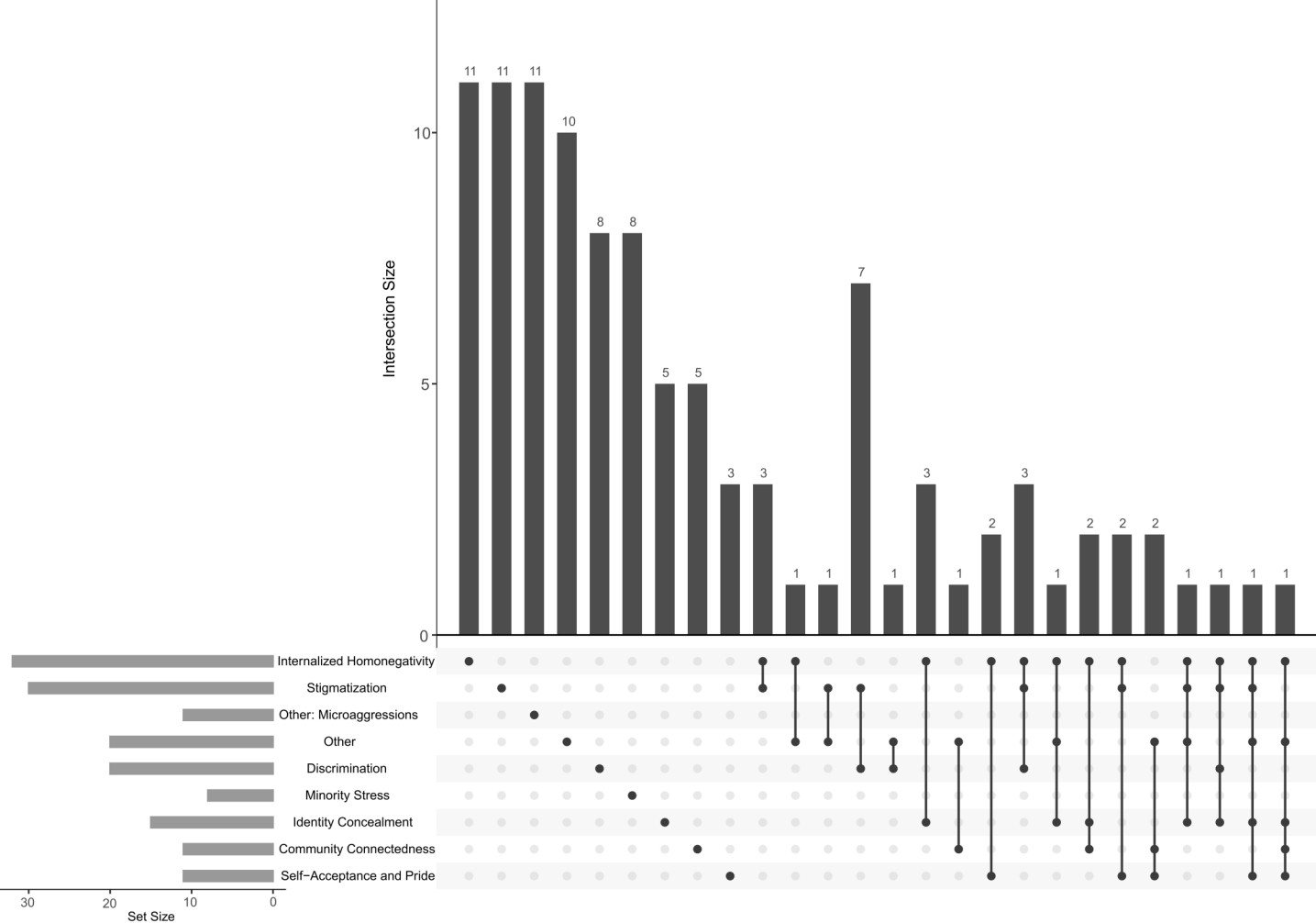

**Fig 3. Construct coverage across all included measurement instruments.** Horizontal bars (Set Size, bottom left) indicate how many measurement instruments assess each construct, regardless of whether they assess additional constructs. Vertical bars (Intersection Size, top) show how many measurement instruments assess the exact combination of constructs marked by filled dots below. Single-dot bars reflect measurement instruments measuring one construct only, while multiple dots indicate measurement instruments measuring that specific combination of constructs.

construct. 64 instruments focus on individual dimensions of minority stress as single constructs, such as internalized homonegativity or stigmatization. In contrast, 33 instruments capture multiple minority stress dimensions, indicating that multidimensional assessment remains comparatively rare.

The following eight instruments explicitly aim to measure minority stress:

1. The Daily Heterosexist Experiences Questionnaire (DHEQ) [64–66]

2. Daily Sexual Minority Stressors Scale (DSMSS) [171,172]

3. Emerging Adult Inventory of Minority Stress (EAIMS) [73]

4. LGBT Minority Stress Measure [68,69]

5. Military Minority Stress Scale (MMSS) [72]

6. Minority Stress Scale (MSS) [70,71]

7. Concealment stress during adolescence and Sexual minority-related positive identity development (SM-PID) [164]

8. Sexual Minority Stress Scale (SMSS) [67]

Eleven instruments exclusively assess internalized homonegativity. Following Meyer's conceptualization [11], this construct represents a proximal minority stressor, reflecting the internalization of negative societal attitudes toward one's sexual identity. This applies to the following instruments:

1. Self-Stigma Scale–Short Form (SSS–S) [105]

2. The Wright Internalized Homophobia Scale [111]

3. Internalized Homophobia Scale (IHS) [93,112–114]

4. The Short Internalized Homonegativity Scale (SIHS) [136,137]

5. Internalized Homophobia (IH) Scale [31,142]

6. The Reactions to Homosexuality Scale (RHS) [143–145]

7. Multi-Axial Gay Men's Inventory-Men's Short Version (MAGI-MSV) [151,152]

8. Internalized Homophobia Scale for Gay Chinese Men [156]

9. Internalized Sexual Prejudice Scale (ISPS) [158]

10. Internalized Bi/Homophobia Scale (IBHS) [169]

11. The Internalized Homophobia Scale for Vietnamese Sexual Minority Women (IHVN-W) [178]

Eleven instruments focus exclusively on stigmatization-related processes. In minority stress theory, perceived stigma and expectations of rejection are conceptualized as proximal stressors that arise from the awareness of prevailing societal stigma [11]. Instruments in this category assess vigilance, anticipated discrimination, or sensitivity to rejection based on sexual identity, without addressing internalized stigma or concealment. The following measurement instruments fall into this category:

1. Discrimination-Related Vigilance Scale (DRVS) [94]

2. LGBTQ-Hypervigilance Scale [38]

3. Fear of Heterosexism Scale (FoHS) [103]

4. Revised Internalized Homophobia (IHP-R) Scale [15,104]

5. Stigma-Consciousness Questionnaire (SCQ) for Gay Men and Lesbians [106–108]

6. Perceived Familial Stigma of Sexuality Questionnaire (PFSSQ) [109]

7. Gay-Related Rejection Sensitivity (RS) Scale [35,93,133]

8. Perceived Stigma Measure [37]

9. Sexual Behaviour Stigma Scale [160–162]

10. Anticipated stigma scale [170]

11. Sexual Minority Women-Rejection Sensitivity Scale (SMW-RSS) [180]

Eleven measurement instruments assess microaggressions, conceptualized as everyday verbal, behavioral, or environmental acts that, independent of intent, convey demeaning or hostile messages to the targeted person or group [208]. The LGBT POC Microaggressions Scale (LGBT-PCMS) [187,188] and its brief version, the LGBTQ+ PCMS-B [189], index intersectional microaggressions among people of color; the Sexual Orientation Microaggression Inventory (SOMI) [190,191] characterizes anti-gay expressions, denial, and societal disapproval; the Sexual Orientation Microaggressions Scale (SOMS) [192–194] maps microinvalidations, heterosexist language, and gender role enforcement; Homonegative Microaggressions Scale (HMS) [195,196] captures homonegative slights and second class treatment; the Bisexual Microaggressions Scale targets binegative erasure [200]; LGBT microaggression experiences at work scale (LGBT-MEWS) [197] focuses on workplace slights; the LGBQ Microaggressions on Campus Scale [198] indexes campus interpersonal and environmental slights; and Thai Sexual Orientation Microaggressions Scale (T-SOMG) [199] provides another measure for microaggressions. Some measurement instruments also capture microaffirmations, that are subtle verbal, behavioral, or environmental signals that convey inclusion, respect, and valuing of marginalized individuals or groups experiences [199]. Notably the Bisexual Microaggression and Microaffirmation Scales for Women (BMMS-W) [201–203] pairs microaggressions with microaffirmations and the Thai Sexual Orientation Microaffirmations Scale (T-SOMF) [199] specifically indexes microaffirmations.

Eight instruments focused specifically on discriminatory stressors. The Heterosexist Harassment, Rejection, and Discrimination Scale (HHRDS) [84–87] assesses heterosexist harassment and rejection across interpersonal and institutional settings, the Everyday Discrimination Scale (EDS) [92–94] indexes routine unfair treatment in daily life, and the Sexual Orientation Experiences of Discrimination (SOEOD-9 and SOEOD-5) [110] capture sexual orientation based discrimination across multiple domains. The Homophobic Bullying Scale (HBS) [117] measures homophobic bullying, the Enacted Sexual Stigma Experiences Scale in Military Service (ESSESiMS) [155] captures enacted sexual stigma in military contexts, and the ABES [182] together with the Brief Anti-Bisexual Experiences Scale (Brief ABES) [183] gauge anti bisexual experiences including instability stereotypes, irresponsibility attributions, and interpersonal hostility.

Five instruments capture concealment and disclosure processes that shape exposure to stressors. The Coming Out Vigilance (COV) Measure [95] assesses vigilance about potential disclosure and its consequences; the Sexual Orientation Concealment Scale (SOCS) [100] quantifies active concealment of sexual orientation; the Lesbian, Gay, Bisexual-Visibility Management Scale (LGB-VMS) [101] assesses strategies for managing visibility across settings; the Nebraska Outness Scale (NOS) [93,102] profiles degree of outness across social domains; and A Lesbian Identity Disclosure Assessment (ALIDA-II) [179] evaluates disclosure within relationships and public contexts.

Five measurement instruments operationalized access to community-level resources relevant to coping with identity-related stressors. The Brief Sense of Community Scale (BSCS) [120] assesses a general sense of community; the Psychological Sense of LGBT Community Scale (PSOC-LGBT) [121] measures psychological sense of community among sexual minorities; the Connectedness to the LGBT Community Scale [43,122] gauges perceived closeness and involvement with LGBT communities; the LGBTQ Belongingness Attainment Scale (LGBTQ BAS) [123] captures belongingness attainment via connectedness, affiliation, and companionship; and the Relational Health Indices–Community (RHI-C) [124] reflects the quality of community relational ties.

Three instruments focus exclusively on positive identity processes: self-acceptance (SASI) [89], facets of positive bisexual identity (PBI) [184], and perceived benefits and strengths associated with non-heterosexual identities (PANQ) [96].

The following instruments capture constructs that do not directly map onto Meyer's minority stress model [11,12,41] but were included due to their conceptual relevance to minority stress experiences. While they do not assess distal or proximal stressors as defined by Meyer [11], they address themes such as intra-minority dynamics, social support, and identity-related cognitions, which may influence or co-occur with minority stress. The measurement instruments focusing exclusively on these constructs are:

1. Sexual Orientation Reflection and Rumination Scale (SRRS) [81]

2. Positive Coming Out Responses (PCOR) Measure [95]

3. Courage to Challenge Scale [128]

4. The Lesbian, Gay, Bisexual and Transgendered Climate Inventory (LGBTCI) [118,119]

5. Multidimensional Scale of Perceived Social Support (MSPSS) [125,126]

6. Index of Sojourner Social Support (ISSS) [166]

7. Network Social Capital Scale [165]

8. Gay Community Stress Scale (GCSS) [129–132]

9. Homophobia Management Scale [167]

10. Gay-Specific Intraminority Stigma Inventory (G-SISI) [168]

**Target populations**

Fig 4 displays the number of measurement instruments by gender and sexual identity.

There's a noticeable emphasis on instruments targeting LGB populations, with 57 instruments (54%), followed by 24 instruments (23%) for lesbian or gay populations. With only six (6%) and five (5%) measurement instruments respectively, lesbian/bisexual (DSMSS [171,172], SMW-RSS [180], Sexual Stigma Scale Adapted for Lesbian, Bisexual and Queer Women [181], Lesbian Internalized Homophobia Scale [LIHS] [174–176], IHVN-W [178], Chinese and Chinese American Sexual Minority Women Internalized Heterosexism [CCSMW-IH] Scale [203]) and exclusively bisexual populations (ABES [182], Brief-ABES [183], BMMS-W [201–203], PBI Scale [184], BII [185], CIBS [186]) are targeted.

Regarding gender, 66 measurement instruments (63%) are validated for both women and men. Thereof, 30 instruments (29%) target only (cis- and/or trans-) men, whereas nine instruments (9%) target only (cis- and/or trans-) women.

Instruments targeting only men (30 instruments):

1. Anticipated Stigma Scale [170]

2. The China MSM Stigma Scale [18]

3. Chinese Homosexuality-Related Stigma Scales [163]

4. Enacted Stigma Scale [170]

5. The Experienced and Anticipated Sexual Stigma Scale in Health-care Services (EASSSiHS) [154]

6. The Enacted Sexual Stigma Experiences Scale in Military Service (ESSESiMS) [155]

7. Experiences and Perceptions of Discrimination Scale [159]

8. The Gay Identity-based Shame and Pride Scales [149]

9. Gay Male Heterophobia Scale [157]

10. Gay-Specific Intraminority Stigma Inventory (G-SISI) [168]

11. Homophobia Management Scale [167]

12. Homosexuality-Related Stigma Scale [146]

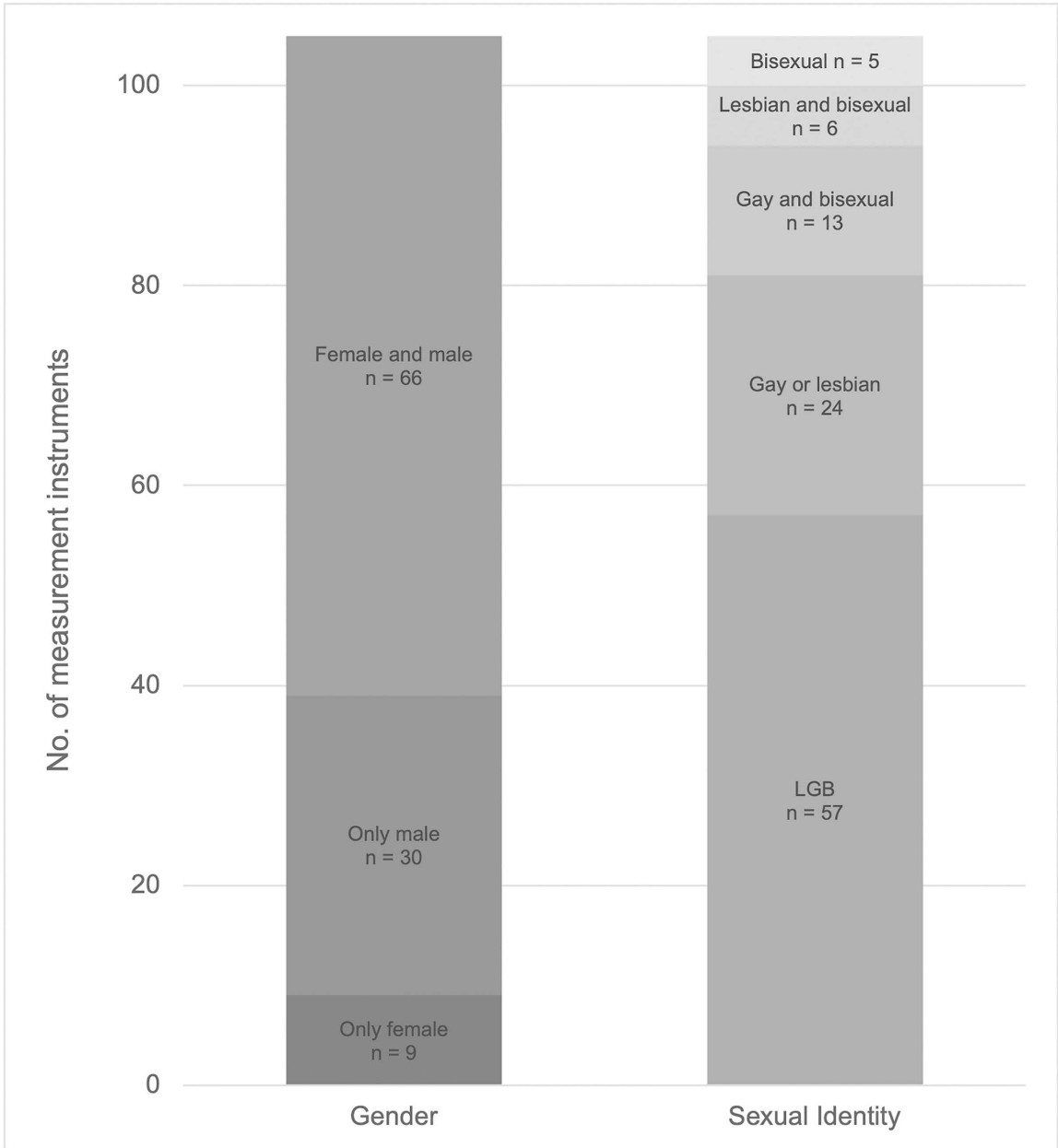

**Fig 4. Number of measurement instruments by gender and sexual identity.**

13. Internalized Bi/Homophobia Scale (IBHS) [169]

14. Internalized Homonegativity Inventory (IHNI) [28,138–140]

15. Internalized Homophobia Measure [37]

16. Internalized Homophobia (IH) Scale [31,142]

17. Internalized Homophobia Scale for Gay Chinese Men [156]

18. Internalized Sexual Prejudice Scale (ISPS) [158]

19. Index of Sojourner Social Support (ISSS) [166]

20. Multi-Axial Gay Men's Inventory-Men's Short Version (MAGI-MSV) [151,152]

21. Multidimensional Sexual Identity Stigma (MSIS) Scale [148]

22. The Neilands Sexual Stigma Scale [153]

23. Network Social Capital Scale [165]

24. Nungesser Homosexual Attitudes Inventory (NHAI) [32,150]

25. Shidlo-revised Nungesser Homosexual Attitudes Inventory (NHAI-SR) [29,150]

26. Perceived Stigma Measure [37]

27. Reactions to Homosexuality Scale (RHS) [143–145]

28. Subjective Scale of Stigma and Discrimination (SISD) [147]

29. Sexual behaviour stigma scale [160–162]

30. Concealment stress during adolescence and Sexual minority-related positive identity development (SM-PID) [164]

Instruments targeting only women (9 instruments):

1. A Lesbian Identity Disclosure Assessment (ALIDA-II) [179]

2. Bisexual Microaggression and Microaffirmation Scales for Women (BMMS-W) [201–203]

3. Daily Sexual Minority Stressors Scale (DSMSS) [171,172]

4. The Internalized Homophobia Scale for Vietnamese Sexual Minority Women (IHVN-W) [178]

5. Lesbian Internalized Homophobia Scale (LIHS) [174–176]

6. The Self-Identified Lesbian Internalized Homophobia Scale (SLIHS) [177,207]

7. Sexual Minority Women-Rejection Sensitivity Scale (SMW-RSS) [180]

8. Sexual Stigma Scale Adapted for Lesbian, Bisexual and Queer Women [181]

9. Chinese and Chinese American Sexual Minority Women Internalized Heterosexism (CCSMW-IH) Scale [173].

## Short versions and adaptations of minority stress measurement instruments

We identified three short versions of measurement instruments: The Short Internalized Homonegativity Scale (SIHS) [136,137], based on the Internalized Homophobia Scale (IH Scale) [31,142], the Brief Anti-Bisexual Experiences Scale (Brief ABES) [183], based on the ABES [182] and The LGBTQ+ POC Microaggressions Scale-Brief (LGBTQ+PCMS-B) [189], based on the LGBT People of Color Microaggressions Scale (LGBT-PCMS) [187].

The Self Stigma Scale – Short Form (SSS-S) [105], despite implying the existence of an original longer version, represents the validated result of a development process that started with a longer instrument. Additionally, while the "Sexual Orientation Experiences of Discrimination-5" (SOEOD-5) is shorter than the SOEOD-9, its development was not intended to create a shorter version. It demonstrated improved psychometric properties and is therefore not considered a short version of the SOEOD-9 [110].

Moreover, numerous adaptations of measurement instruments emerged (excluding language validations), some of which are based on the same original instruments:

1. The Lesbian, Gay, and Bisexual Identity Scale (LGBIS) [42,74–80], based on the Lesbian and Gay Identity Scale (LGIS) [134].

2. The Reactions to Homosexuality Scale (RHS) [143–145], based on the Internalized Homophobia Scale (IH Scale) [31,142].

3. The Revised Internalized Homophobia (IHP-R) Scale [15,104], based on the Internalized Homophobia Scale (IHP).

4. The Internalized Homophobia Scale (IHS) [112–114], also based on the Internalized Homophobia Scale (IHP).

5. The Internalized Bi/Homophobia Scale (IBHS) [169], based on the Internalized Homophobia Scale (IHS) [136,137,142].

6. The Shidlo-revised Nungesser Homosexual Attitudes Inventory (NHAI-SR) [29,150], based on Nungesser Homosexual Attitudes Inventory (NHAI) [32,150].

7. The China MSM Stigma Scale [18], based on a scale by Díaz et al. [206]

8. The Neilands Sexual Stigma Scale [153], based on The China MSM Stigma Scale [18].

The IHP and the scale by Díaz et al. [206] are not included in this review, as no psychometric studies were identified for these scales. Across adapted measures, the most common modifications were item reductions and revisions of measurement constructs. A side-by-side comparison of these adaptations is presented in Supplement S4 File.

Highlighting the field's dynamic evolution, eight measurements instruments predate the last two decades, and therefore Meyer's minority stress model:

1. The Nungesser Homosexual Attitudes Inventory (NHAI) [32]

2. The Shidlo-revised Nungesser Homosexual Attitudes Inventory (NHAI-SR) [29]

3. The Gay Identity-based Shame and Pride Scales [149]

4. The Internalized Homophobia (IH) Scale [31]

5. The Stigma-Consciousness Questionnaire (SCQ) for gay men and lesbians [106]

6. The Lesbian and Gay Identity Scale (LGIS) [134]

7. The Internalized Homonegativity Inventory (IHNI) [28]

8. The Lesbian Internalized Homophobia Scale (LIHS) [174].

When mapped against the year of the first publication of a measurement instrument, Fig 5 illustrates how the development of instruments has shifted across minority stress-related constructs over time. Early measures, appearing in the 1980s and 1990s, predominantly targeted internalized homonegativity and identity concealment, reflecting the initial focus on intrapersonal processes. From the early 2000s onwards, instrument development diversified to include stigmatization, discrimination, and other constructs elaborated above. This diversification accelerated markedly after 2010, with the 2010s and 2020s showing the greatest density of new measures. During this period, stigmatization emerged as the most frequently operationalized construct.

## Diverse cultural and linguistic contexts

While a significant proportion of the research and instrument development is concentrated in Western contexts, particularly in the United States, we have identified instruments that have been validated in a variety of

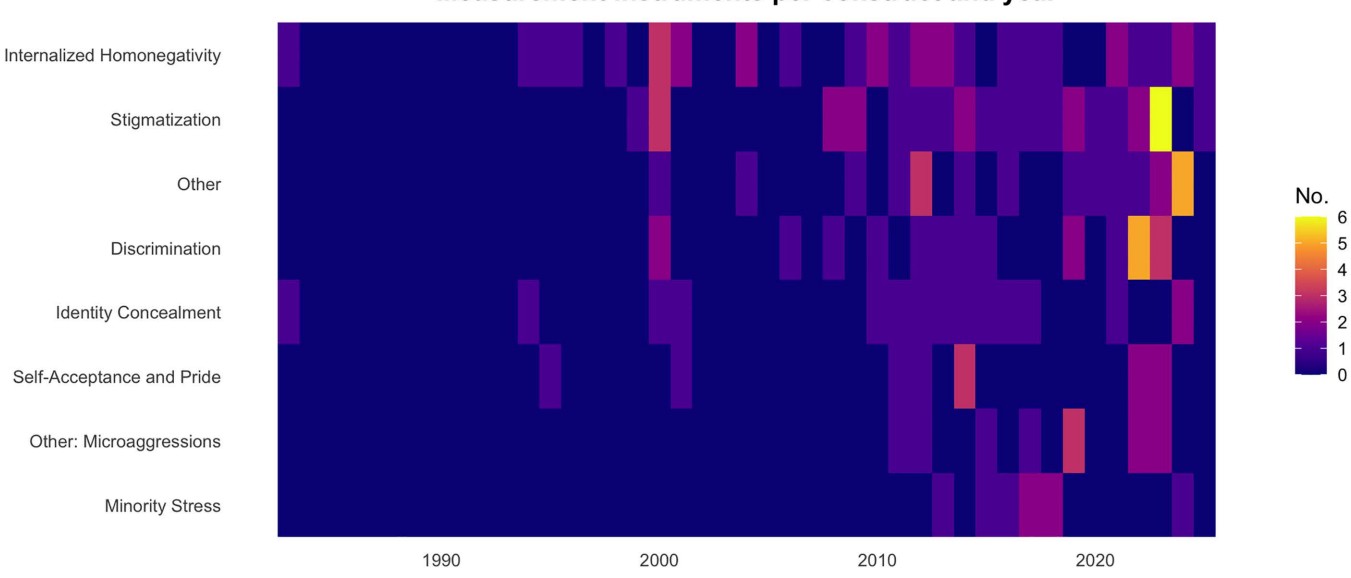

**Fig 5. Heat map of measurement instruments by construct and year.**

languages beyond English. These include Croatian, Polish, Italian, Traditional Chinese, Spanish, Vietnamese, Xhosa, Luganda, German, Turkish, Portuguese, Russian, Mexican Spanish, Thai, Chinese, and Kiswahili [66,71,80,92,98,99,104,107,113,114,119,121,131,132,137,140,142–145,148,153,161,162,175,176,186,191,194]. Specifically, we identified 30 studies including language validations. Notably, the RHS [142–145] was translated into at least 25 languages which positions the RHS as the instrument with the highest number of language validations in our review. Additionally, 17 measurement instruments were originally developed in a language other than English [67,70,82,105,116,135,146,147,156,163,166,178,185,199]. The LGBTCI [118,119] and the LGBT-MEWS [197] were identified as tools developed with a focus on the work environment context. These instruments assess factors related to the workplace experiences of LGBT employees, such as discrimination, acceptance, and support. Additionally, the HHRDS [84–87] encompasses aspects related to workplace and school discrimination, rendering it pertinent in a work environment context. The LGBQ Microaggressions on Campus Scale [198] is designed to assess microaggressions in a college setting. The POHS [88] focuses specifically on experiences in an online setting, while the EASSSiHS [154] assesses stigma in the health care sector. Furthermore, the MMSS [72] and the ESSESiMS [155] specifically address (active or former) military personnel and the LGBIS was additionally validated in military veterans [75]. Intraminority stress is assessed by two measurement instruments: the GCSS [129–132] and the G-SISI [168].

Our review identified nine instruments (The Internalized Heterosexist Racism Measure [IHRM] [115], HHRDS [85], LGBT-PCMS [187,188], LGBTQ+ PCMS-B [189], RHI-C [124], MSPSS [125,126], Enacted Stigma Scale and Anticipated Stigma Scale [170], Connectedness to the LGBT Community Scale [122] and BMMS-W [202,203]) that have been specifically validated for Black and Indigenous people and people of color (BIPOC). This may suggest an emerging focus on intersectional identities within sexual minority stress research.

## Discussion

Our scoping review has identified an abundance of measurement instruments, surpassing initial expectations of a limited number of measurement instruments in the field of minority stress. This array of instruments highlights the complexity and

depth of research in the field of sexual minority stress. However, our review cannot determine which measurement instrument is most suitable for addressing research questions related to minority identity in terms of psychometric properties. Consistent with PRISMA-ScR guidance [60], no risk-of-bias assessment was conducted for this scoping review.

We observed that there are numerous measurement instruments available for internalized homonegativity and stigmatization. However, aligning the name and construct of the measurement instruments proved challenging with the risk of falling into the jingle-jangle fallacy. The importance of meticulous examination and comparison of measurement tools became evident. Further, our review revealed a predominant research focus on men in the existing literature. In contrast, instruments targeting women and bisexual individuals, are relatively rare. Most instruments were developed or validated in the United States and Europe, that is, predominantly high-income settings.

These findings support the objective of guiding clinicians and researchers in selecting appropriate measurement instruments. Consistent with the broader role of scoping reviews in identifying conceptual and methodological gaps in complex fields [56,57,209] this review offers a structured overview of current measurement instruments and highlights priorities for future research. By mapping available measures, the review advances efforts to harmonize construct definitions in an unequivocal way [54], promoting a more coherent and inclusive approach to capturing sexual identity-related experiences.

### Relevant sources for minority stress measurement instruments

Notably, a significant proportion of measurement instruments were extracted from grey literature sources, particularly dissertations. This challenges prevailing guidelines for systematic literature searches concerning measurement instruments. Recommendations suggest a low probability of encountering additional relevant articles for reviews of Patient-Reported Outcome Measures (PROMs) within supplementary sources such as grey literature [210]. However, our review process has yielded a noteworthy quantity of measurement instruments from these sources. Monographs are often difficult to access, and it remains unclear whether any relevant measurement instruments of high quality are indeed overlooked. It should also be noted that the research focus of this scoping review is not on PROMs but on outcome measurement instruments outside of clinical research. Therefore, the observations made here may not apply to PROMs.

### Focus on positive vs. negative aspects of minority identity

The scoping review has identified a number of instruments addressing the adverse components of minority stress, such as internalized homonegativity and stigmatization. Contrastingly, positive facets of minority identity, specifically self-acceptance and pride, are less represented in the literature but acknowledged as areas requiring further investigation [164,211,212]. This pattern should be interpreted with caution: although our search combined broad population and instrument terms to capture sexual-identity minority stress measures (see Supplement S2 File), we did not explicitly include keywords for positive identity constructs, which may have limited retrieval. Nonetheless, within the included set, our temporal mapping (Fig 4) shows that earlier instruments targeted negative constructs (e.g., internalized homonegativity, stigmatization, identity concealment), with positive constructs (e.g., self-acceptance and pride) emerging later. Furthermore, this observation aligns with Meyer's call for stronger integration of resilience within health research on LGBT populations [41]. The minority strength framework proposed by Perrin et al. [211] suggests that factors such as identity pride and community connectivity can mediate favorable health outcomes in LGBTQ populations. Therefore, a comprehensive understanding of the LGBQ experience thus necessitates the inclusion of both stress and resource-based perspectives [11,41,213].

### Implications of jingle-jangle fallacies

Upon analysis of the range of constructs captured by the measurement instruments, a noteworthy observation emerged regarding their naming conventions. It was evident that some instruments, despite having similar names, assess different constructs or delve into varying depths within a construct. For instance, "The Daily Heterosexist Experiences

Questionnaire (DHEQ)" [64] and the "Anti-Bisexual Experiences Scale (ABES)" [182] may serve as an example, as both questionnaires are, by name, directed at negative experiences. Upon closer inspection, it becomes evident that the DHEQ addresses the broad construct of minority stress, encompassing nine subscales. In contrast, the ABES specifically targets anti-bisexual experiences, focusing primarily on proximal factors of minority stress. This scenario could be indicative of a jingle fallacy, where distinct constructs or concepts are erroneously assumed to be identical due to similar nomenclature [54,55].

Conversely, we observed instances where instruments with different names investigate similar or closely related constructs. For example, the "Multifactor Internalized Homophobia Inventory (MIHI)" [135] assesses not only internalized homophobia but also other facets of minority stress such as community connectedness and outness. In this, the MIHI demonstrates parallels to the "Sexual Minority Stress Scale (SMSS)" [67]. This exemplifies a potential jangle fallacy, where similar constructs are wrongly considered distinct due to different labels [54,55].

The challenges in clearly delineating these constructs are highlighted by the fact that the review by Fisher et al. [214] on microaggression measurement tools included several instruments that were also incorporated in this review for constructs other than microaggressions.

Jingle-jangle fallacies pose a challenge, as they can lead to confusion and misinterpretation of constructs, hindering a clear understanding of minority stress dynamics while also impeding communication between researchers and affected persons. The jingle fallacy, in particular, risks oversimplifying the complexity of experiences by merging different constructs under a single label. In contrast, the jangle fallacy might contribute to unnecessary fragmentation in research, concealing the similarities between closely related constructs. This observation warrants attention, given the current lack of discourse surrounding the jingle-jangle fallacy in the field of minority stress and LGBQ health research.

Our findings underscore the need for further investigation into the reasons behind and barriers to the lack of standardization in this field. Identifying these reasons and barriers is crucial for developing a cohesive framework for measuring minority stress. As seen in other fields, such as discrimination measurement [53], rigorous construct definitions are essential for addressing these challenges. Future research should focus on exploring the multifaceted nature of minority stress, the evolving terminology related to sexual identity experiences, and the interdisciplinary gaps that hinder the development of a standardized approach.

## Gender disparity in instrument focus

A notable trend observed is the strong concentration on men within these instruments. While this may reflect the historical context of the field [1], it highlights a significant gap in research tools specifically addressing the experiences of women, trans and non-binary individuals within the LGBQ community. This disparity necessitates a shift in research focus to encompass a more inclusive understanding of health across all genders within the LGBQ community. Furthermore, Fisher's review of microaggression measurement tools [214], underscores the necessity for diversifying measurement approaches in order to guarantee the emphasizes of tools for specific LGBTQ subpopulations, particularly those with intersecting minority statuses.

## Implications for practice

This scoping review on measurement instruments can guide clinicians and researchers alike to select measurement instruments that align with their intended target construct, population, setting, and language. Advancing measurement in this field requires clearer, more transparent construct definitions for minority stress. Research on LGBQ health has attracted increasing interdisciplinary attention, drawing on theoretical frameworks beyond minority stress theory and thereby broadening the conceptual and methodological foundations for measurement. One example is research on microaggression stemming from critical race theory [215], first applied to sexual identity through measures for people of color, namely the LGBT People of Color Microaggressions Scale (LGBT-PCMS) in 2011 [187], and subsequently applied solely

to sexual-identity minority populations via multiple scales (SOMI [190,191], SOMS [192–194], HMS [195,196], LGBT-MEWS [197], the LGBQ Microaggressions on Campus Scale [198], T-SOMG [199], T-SOMF [199], the Bisexual Microaggressions Scale [200] and the BMMS-W [201]).

Another example is the Network Social Capital Scale which examines social capital among gay and bisexual men who have sex with men using social capital theory, which is notably reported in literature as lacking a definition for social capital [216]. One approach conceptualizes social capital "as the resources […] available to members of social groups" [216, p. 3]. This definition closely aligns with resilience as a resource in the minority stress model, where Meyer defines "community-level resilience [as] tangible and intangible resources in the community" [41, p. 211]. Furthermore, extensive research has advanced Meyer's minority stress model [11] itself, while a unified framework has yet to be established [217].

However, as others have noted, there is no "correct" or "incorrect" construct definition, because psychological constructs are not discrete, naturally bounded entities. Moreover, theoretical and definitional diversity may contribute more to scientific progress than uniformity [54]. Yet, divergent approaches and terminology make it difficult to identify conceptually similar efforts, contributing to the proliferation of instruments observed in this review. In line with Bringmann et al. [218], we therefore encourage researchers to be comprehensive and explicit in deciding on their construct definition, to minimize jingle-jangle fallacies.

A practical route is the decentralized construct taxonomy (DCT) specification by Peters and Crutzen [54]. A DCT specification consists of (1) instructions to measure the construct; (2) classification of existing instruments measuring the construct; (3) instructions for qualitative research that would investigate the construct; and (4) instructions for coding of qualitative data relevant to the construct's content. Mapping measurement instruments to a DCT specification would link instruments with similar constructs but also clarify differences which therefore need to be interpreted distinctly. Such an effort would be an important step towards harmonization of the measurement field around sexual identity minority stress.

## Limitations

Considering the inherent nature of a scoping review, which aims to map the existing literature rather than assess the quality of studies, our work does not conclude on the quality of the psychometric studies or the validity of the instruments. A dedicated systematic review is therefore warranted to conduct a formal psychometric evaluation, including risk-of-bias assessment and grading of certainty of evidence, to identify instruments best suited for research and practice.

Despite an extensive search and inclusive selection criteria, this review does not claim to be exhaustive. Only studies involving the development or validation of instruments were included. This criterion led to the exclusion of the original "Internalized Homophobia Scale (IHP)" by John Martin, which lacked accessible psychometric studies. The decision to omit instruments without clear validation was taken to ensure the measurement instruments met a minimum standard of quality that would render them applicable in minority stress research.

Furthermore, the scenario where Herek developed two scales, the IHS [112] and the IHP-R [15], both adaptations of the original IHP, exemplifies the challenge of delineating constructs within our review. Despite originating from the same scale, these instruments explore different constructs – stigmatization and homophobia – highlighting the difficulty of categorizing measurement instruments. This situation serves further as a testament to the vast array of instruments identified. In light of these challenges, we relied on the constructs as defined by the developers of the instruments for categorization. It is important to acknowledge, however, that a more in-depth investigation might reveal alternative categorizations.

## Conclusion

In conclusion, this review shows the vast diversity of instruments developed to measure sexual minority stress, while also identifying gaps and biases in the current research landscape. The emergence of instruments from grey literature and the lack of intersectionality mark significant strides in the field. However, the need for more gender-balanced research tools and a broader exploration of both positive and negative aspects of minority identity is evident. It is therefore essential to

integrate a critical examination of the jingle-jangle fallacies within this context, as it offers a pathway to refine measurement approaches and enhance the validity of research findings. As the field continues to evolve, it is imperative to address these gaps to ensure a holistic understanding of the LGBQ experience as well as a thorough systematic analysis of the measurement properties of instruments focusing on the comprehensive concept of minority stress.

## Supporting information

**S1 File. Supplement 1.** PRISMA-ScR Checklist.
(DOCX)

**S2 File. Supplement 2.** Search Strategy.
(DOCX)

**S3 File. Supplement 3.** Data extraction form.
(DOCX)

**S4 File. Supplement 4.** Adaptions of measurement instruments.
(DOCX)

## Positionality statement

All independent reviewers involved in screening and data extraction are researchers trained in evidence synthesis with disciplinary backgrounds in psychology, public health, and epidemiology, working within Western European public institutions. We approach the topic as scholarly allies rather than community representatives and do not claim lived-experience authority. We recognize that our personal background as well as our disciplinary training, institutional context, and the English/German scholarly lens can shape interpretation. We therefore employed rigorous methodology and reflected on potential normative assumptions and made the fewest possible adjustments to the studies' descriptions included and documented any necessary clarifications transparently (see Data Analysis and table footnotes).

## Acknowledgments

We thank Uta Merkel for her review of the search strategy and constructive feedback.

## Author contributions

**Conceptualization:** Maria Misevic-Kallenbach, Petra Warschburger.

**Data curation:** Jacqueline Schirm.

**Formal analysis:** Maria Misevic-Kallenbach, Susann Conrad, Simone Freitag, Anja Jacobs, Madlen Sixtensson.

**Investigation:** Maria Misevic-Kallenbach, Susann Conrad, Simone Freitag, Anja Jacobs, Jacqueline Schirm, Madlen Sixtensson.

**Methodology:** Maria Misevic-Kallenbach, Susann Conrad, Simone Freitag, Anja Jacobs, Jacqueline Schirm, Madlen Sixtensson.

**Supervision:** Petra Warschburger.

**Validation:** Susann Conrad, Simone Freitag, Anja Jacobs, Madlen Sixtensson.

**Visualization:** Maria Misevic-Kallenbach.

**Writing – original draft:** Maria Misevic-Kallenbach.

**Writing – review & editing:** Susann Conrad, Simone Freitag, Anja Jacobs, Madlen Sixtensson, Petra Warschburger.

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
