## [Decision Letter · Decision Letter 0]

15 Sep 2025

Dear Dr. Misevic-Kallenbach,

Thank you for submitting your manuscript to PLOS ONE. After careful consideration, we feel that it has merit but does not fully meet PLOS ONE’s publication criteria as it currently stands. Therefore, we invite you to submit a revised version of the manuscript that addresses the points raised during the review process.

I appreciate the authors for this manuscript which attempts to address an important and timely topic, and a comprehensive review of this nature is indeed much needed. However, the current version requires substantial revisions to enhance its clarity, depth, and overall contribution to the literature.

The introduction should be expanded to provide a clearer framing of the study. Specifically, the statement “However, such a focus potentially neglects the multifaceted experiences of LGBTQ individuals, extending beyond sexual health inequalities” needs elaboration. The authors should clarify what is meant by “neglect” and specify the kinds of inequalities or experiences that have been overlooked in prior work. This will help set a stronger rationale for the review.

Meyer’s minority stress model is mentioned very superficially but not well discussed. The literature review should engage more deeply with this model, explaining how minority stress has been conceptualized in earlier studies and synthesizing the evidence around it. This should then be linked explicitly to the gap that this review is aiming to address.

The methods section requires more detail with better transparency and reproducibility. For example, the authors should describe all components of the process, including what the data extraction sheet looked like (attachment is needed), how data extraction was performed, whether a risk of bias assessment was conducted (if not why), and how decisions were made during review. These are very important elements that currently appear to be missing.

The results section is underdeveloped and should provide a richer unpacking of the findings. It should discuss the constructs studied, the contexts in which they were applied, and whether their validity and reliability were assessed or established using validated scales. If adaptations of constructs were made in the included studies, the authors should describe these adaptations and discuss any evidence of limitations or challenges in their use.

Similarly, the discussion section should be substantially strengthened. It should clearly articulate what this review contributes to the field and how it advances current knowledge. At present, the discussion feels cursory and does not fully capitalize on the findings. Lessons learned would be better integrated into the results or discussion sections, rather than being presented in the introduction.

Finally, the paper would benefit from a thorough reflection on the validity and reliability of the constructs examined across the reviewed studies, as this is central to the credibility of the conclusions drawn.

We look forward to receiving your revised manuscript.

Kind regards,

Anupam Joya Sharma

Guest Editor

PLOS ONE

Journal Requirements:

3. Please include captions for your Supporting Information files at the end of your manuscript, and update any in-text citations to match accordingly. Please see our Supporting Information guidelines for more information: http://journals.plos.org/plosone/s/supporting-information .

Additional Editor Comments (if provided):

I appreciate the authors for this manuscript which attempts to address an important and timely topic, and a comprehensive review of this nature is indeed much needed. However, the current version requires substantial revisions to enhance its clarity, depth, and overall contribution to the literature.

The introduction should be expanded to provide a clearer framing of the study. Specifically, the statement “However, such a focus potentially neglects the multifaceted experiences of LGBTQ individuals, extending beyond sexual health inequalities” needs elaboration. The authors should clarify what is meant by “neglect” and specify the kinds of inequalities or experiences that have been overlooked in prior work. This will help set a stronger rationale for the review.

Meyer’s minority stress model is mentioned very superficially but not well discussed. The literature review should engage more deeply with this model, explaining how minority stress has been conceptualized in earlier studies and synthesizing the evidence around it. This should then be linked explicitly to the gap that this review is aiming to address.

The methods section requires more detail with better transparency and reproducibility. For example, the authors should describe all components of the process, including what the data extraction sheet looked like (attachment is needed), how data extraction was performed, whether a risk of bias assessment was conducted (if not why), and how decisions were made during review. These are very important elements that currently appear to be missing.

The results section is underdeveloped and should provide a richer unpacking of the findings. It should discuss the constructs studied, the contexts in which they were applied, and whether their validity and reliability were assessed or established using validated scales. If adaptations of constructs were made in the included studies, the authors should describe these adaptations and discuss any evidence of limitations or challenges in their use.

Similarly, the discussion section should be substantially strengthened. It should clearly articulate what this review contributes to the field and how it advances current knowledge. At present, the discussion feels cursory and does not fully capitalize on the findings. Lessons learned would be better integrated into the results or discussion sections, rather than being presented in the introduction.

Finally, the paper would benefit from a thorough reflection on the validity and reliability of the constructs examined across the reviewed studies, as this is central to the credibility of the conclusions drawn.

While the topic is important and this review has the potential to be a valuable contribution, substantial revisions are needed to deepen the analysis, clarify the methodology, and strengthen the interpretation of results.

Reviewers' comments:

Reviewer's Responses to Questions

**Comments to the Author**

1. Is the manuscript technically sound, and do the data support the conclusions?

Reviewer #1: Yes

Reviewer #2: Partly

2. Has the statistical analysis been performed appropriately and rigorously?

Reviewer #1: Yes

Reviewer #2: Yes

3. Have the authors made all data underlying the findings in their manuscript fully available?

Reviewer #1: Yes

Reviewer #2: Yes

4. Is the manuscript presented in an intelligible fashion and written in standard English?

Reviewer #1: Yes

Reviewer #2: Yes

Reviewer #1: The purpose of this paper was to systematically map measurement instruments for sexual minority stress among LGBQ populations. The focus of the study fits within the scope of the journal and is of relevance to its readership. Below, I note specific areas to strengthen the manuscript and note concerns regarding implications of the study findings. I wish the authors the best of luck as they continue to move this research forward.

Introduction

- Overall, the Introduction is brief and cursory. It would benefit from additional details about the different constructs within the Minority Stress model and how those constructs may be measured to further elaborate the need for the scoping review. This is particularly important given that the Results are organized by constructs.

- It is unclear why the “Lessons Learned” section is in the Introduction as this is better suited for the Discussion/Conclusion.

Methods

- More information regarding the independent reviewers for the stages of screening would be helpful – are their any parts of their background (e.g., academic, personal) that would influence screening? For example, anything regarding reflexivity/positionality.

- Details regarding the consensus process would also be helpful. Who resolved conflicts?

- Can the authors please provide the data extraction form?

Results

- There are a series of measures (page 10-11, lines 214-238) that are listed at bullet points. It may be more appropriate to describe these in a narrative format and provide additional details regarding the constructs they address.

- Reference(s) are needed to support the claim made on lines 240-243.

- For the measures listed as having been adapted (page 14, line 328), it would be helpful to know what the scope of the adaptation(s) were.

Discussion

- Line 384: study is described as a systematic literature review, but was initially framed as a scoping review.

- The authors state that there were limited instruments for assessing positive aspects of minority identity, but it does not appear as though these types of measures were explicitly sought out for based on the search strategy.

- It is not until the Discussion that we know that Dissertations were included. Typically, peer-reviewed sources are only included. Rationale for including Dissertations is necessary as this can limit the rigor/validity of the review.

- For the section beginning on line 411, I again raise the concern of positive aspects of minority identity not being included in the search, thus statements about the limited number of these types of measures should be tempered.

- No implications for practice are presented. It is unclear what the findings of the scoping review contribute to the field and/or advance measurement of minority stress.

Reviewer #2: Review Reports

Title:Measurement instruments for sexual identity minority stress in adults: a scoping review

Review Comments

1.Abstract: Needs further enrichment E.g. stigma is missed, T is missed.

2.Introduction:Is weak for its contents E.g What are the factors associated yet and initiaties for preention.

-E.g2. line 53-54 lacks references

3.Methods

-Sexual identity minority stres should be operationalzed.

-Inclusion criteria is not complete E.g. Language, and year of publication.

-What does 'validated studies' mean? How was the process? What is the justification behind it?

-What does language validation Vs inclusion of other languages like Polish and Italian means?

4.Results and the consequent sections

-Lacks time sequence or evolution

-Contextualization E.g. Countries of.lowincome, middle income and high income.

-Inconsistency E.g. LGBQ Vs LGBTQ, absence of stigma in some sections

-Discussion need more references and explanation.

-THE reference cited should be references

Regards,

**Do you want your identity to be public for this peer review?** For information about this choice, including consent withdrawal, please see our Privacy Policy

Reviewer #1: No

Reviewer #2: No

---

## [Author Response · Author response to Decision Letter 1]

31 Oct 2025

Point-by-point responses to the editors’ and reviewers’ comments

Editor, Anupam Joya Sharma

I appreciate the authors for this manuscript which attempts to address an important and timely topic, and a comprehensive review of this nature is indeed much needed. However, the current version requires substantial revisions to enhance its clarity, depth, and overall contribution to the literature.

Thank you very much for the positive feedback. We carefully considered each comment and improved the manuscript.

Introduction

The introduction should be expanded to provide a clearer framing of the study. Specifically, the statement “However, such a focus potentially neglects the multifaceted experiences of LGBTQ individuals, extending beyond sexual health inequalities” needs elaboration. The authors should clarify what is meant by “neglect” and specify the kinds of inequalities or experiences that have been overlooked in prior work. This will help set a stronger rationale for the review.

Thank you for your suggestion. We expanded the introduction to clarify what we mean by “neglect” and to specify overlooked domains, while keeping the section concise. The revised passage now reads (l. 45-53):

The pursuit of equal rights by Lesbian, Gay, Bisexual, and Queer (LGBQ) activists has raised awareness of health disparities within this community, initially emphasized on the sexual health of gay men due to the HIV/AIDS epidemic (1). However, such a focus potentially neglects the multifaceted experiences of LGBQ individuals, extending beyond sexual health inequalities. LGBQ individuals face unique challenges such as structural stigma (2, 3), barriers to care (4, 5), and discrimination and violence because of their sexual orientation (6). Current research increasingly reveals these challenges and distinct health concerns within this group, including higher rates of depression, anxiety, substance abuse disorders, and suicidal tendencies compared to their heterosexual counterparts (7–10).

Meyer’s minority stress model is mentioned very superficially but not well discussed. The literature review should engage more deeply with this model, explaining how minority stress has been conceptualized in earlier studies and synthesizing the evidence around it. This should then be linked explicitly to the gap that this review is aiming to address.

Thank you for this helpful guidance. We substantially revised the introduction to provide a structured account of Meyer’s model, detailing distal, proximal and resilience processes and their mechanisms, and summarizing empirical support. We also added a concise synthesis of influential extensions and related frameworks to clarify mechanisms and link this overview to the measurement gap that the scoping review addresses. The minority stress model figure is moved to the introduction as Figure 1 to support reader orientation; the PRISMA flow remains in the methods section (p. 3ff).

Methods

The methods section requires more detail with better transparency and reproducibility. For example, the authors should describe all components of the process, including what the data extraction sheet looked like (attachment is needed), how data extraction was performed, whether a risk of bias assessment was conducted (if not why), and how decisions were made during review. These are very important elements that currently appear to be missing.

Thank you for this suggestion. We've updated the methods section to better explain our selection process and provide the data extraction sheets as Supplement 3. Also, we've integrated a statement on positionality at the end of the review (l. 711-720).

Consistent with PRISMA-ScR guidance, our objective was to map measurement instruments rather than to appraise study quality. Therefore, a formal risk-of-bias assessment was beyond the scope of this review. We clarified this point in our Discussion (l. 557-558):

Consistent with PRISMA-ScR guidance (60), no risk-of-bias assessment was conducted for this scoping review.

Results

The results section is underdeveloped and should provide a richer unpacking of the findings. It should discuss the constructs studied, the contexts in which they were applied, and whether their validity and reliability were assessed or established using validated scales.

Thank you for the constructive feedback. We’ve updated the Results to elaborate more on the constructs of the measurement instruments that were previously only listed as bullet points (p. 15f).

Thank you for the suggestion on contextualization. We now report, for each psychometric study, the place and setting of data collection in Table 1 (previously omitted for brevity) to make the development/validation context explicit. Because psychometric studies rarely specify the intended context of use for the measurement instruments, we document the empirical settings as the best available proxy. Most instruments were developed or validated in the United States and Europe (predominantly high-income settings), and we note this concentration and its implications in the Discussion (l. 564-565):

Most instruments were developed or validated in the United States and Europe, that is, predominantly high-income settings.

Thank you also for your comment on validity and reliability. We agree that a more detailed discussion of validity and reliability would be informative to our readers. However, as a scoping review whose aim is to provide an overview of all available instruments, it is not an objective of this work to include only studies with established psychometrics or to synthesize measurement properties. Furthermore, the reporting and assessment of validity and reliability in the included studies is highly heterogeneous and most informative when accompanied by a formal assessment of the risk of bias in the underlying evidence - an undertaking that requires a dedicated systematic review. We have highlighted the need for such a systematic review of measurement properties (including assessment of bias potential) as a clear priority for future work (please see below).

If adaptations of constructs were made in the included studies, the authors should describe these adaptations and discuss any evidence of limitations or challenges in their use.

Thank you for your comment. We’ve updated this section, also in response to the suggestions of reviewer 1, and included a table placing original instruments alongside their adaptations in Supplement 4. The revised text in the manuscript now reads (l. 500-502):

Across adapted measures, the most common modifications were item reductions and revisions of measurement constructs. A side-by-side comparison of these adaptations is presented in Supplement 4.

Consistent with the remit of a scoping review, we did not undertake instrument-by-instrument evaluations of the psychometric properties for adapted (or original) measures; this is outside the scope of this review and would require a systematic evaluation, as is characteristic of systematic reviews. We indicated this as a priority for future work (see the comment after the next one).

Discussion

Similarly, the discussion section should be substantially strengthened. It should clearly articulate what this review contributes to the field and how it advances current knowledge. At present, the discussion feels cursory and does not fully capitalize on the findings. Lessons learned would be better integrated into the results or discussion sections, rather than being presented in the introduction.

Thank you for these suggestions. We’ve substantially rewritten the discussion to meet these points. “Lessons learned” are now integrated in the discussion (l. 552-564) and we’ve included a new section “Implications for practice” to better illustrate the practical relevance of our findings. (p. 26f). In brief, the new section explains that our scoping review helps clinicians and researchers select measures aligned with their target construct, population, setting, and language. It underscores the need for clearer, transparent construct definitions as measurement in LGBQ health increasingly draws on frameworks beyond minority stress theory (e.g., microaggressions, social capital). To reduce jingle-jangle fallacies and improve comparability, we encourage explicit construct specification and propose using decentralized construct taxonomy (DCT) specifications, proposed by Peters and Crutzen (54) to map instruments to constructs, link conceptually similar tools, and clarify non-equivalences, thereby advancing harmonization of measurement in sexual minority stress research.

Finally, the paper would benefit from a thorough reflection on the validity and reliability of the constructs examined across the reviewed studies, as this is central to the credibility of the conclusions drawn.

Thank you for this important point. We agree that a comprehensive appraisal of the validity and reliability of measurement instruments is critical. However, such an assessment lies beyond the scope and intent of this scoping review. Given the methodological heterogeneity of psychometric investigations and the number of instruments, the appropriate next step is a dedicated systematic review with formal appraisal of evidence, including risk-of-bias assessment. We now state this explicitly in the Discussion as a priority for future work (l. 683-685).

A dedicated systematic review is therefore warranted to conduct a formal psychometric evaluation, including risk-of-bias assessment and grading of certainty of evidence, to identify instruments best suited for research and practice.

While the topic is important and this review has the potential to be a valuable contribution, substantial revisions are needed to deepen the analysis, clarify the methodology, and strengthen the interpretation of results.

Thank you for the constructive guidance and the opportunity to revise. We have substantially updated the manuscript to deepen the analysis, clarify the methodology, and strengthen the interpretation of results, and we hope the revisions align with PLOS ONE’s expectations.

Reviewer 1

The purpose of this paper was to systematically map measurement instruments for sexual minority stress among LGBQ populations. The focus of the study fits within the scope of the journal and is of relevance to its readership. Below, I note specific areas to strengthen the manuscript and note concerns regarding implications of the study findings. I wish the authors the best of luck as they continue to move this research forward.

Thank you very much for the positive feedback.

Introduction

1. Overall, the Introduction is brief and cursory. It would benefit from additional details about the different constructs within the Minority Stress model and how those constructs may be measured to further elaborate the need for the scoping review. This is particularly important given that the Results are organized by constructs.

Thank you for your comment. We improved the introduction and expanded the elaboration of the factors included in the Minority Stress Model. To better explain the central constructs, we now clearly define distal stressors, proximal stressors, and resources within the model. We have marked the terms in italics that are central to the results section to provide our readers with more guidance (p. 3ff).

2. It is unclear why the “Lessons Learned” section is in the Introduction as this is better suited for the Discussion/Conclusion.

Thank you for this helpful suggestion. We agree that a “Lessons learned” section is more appropriate for the Discussion. Accordingly, we removed it from the Introduction and integrated the content into the Discussion (l. 552-564). The revised passage now reads:

Our scoping review has identified an abundance of measurement instruments, surpassing initial expectations of a limited number of measurement instruments in the field of minority stress. This array of instruments highlights the complexity and depth of research in the field of sexual minority stress. However, our review cannot determine which measurement instrument is most suitable for addressing research questions related to minority identity in terms of psychometric properties. Consistent with PRISMA-ScR guidance (60), no risk-of-bias assessment was conducted for this scoping review.

We observed that there are numerous measurement instruments available for internalized homonegativity and stigmatization. However, aligning the name and construct of the measurement instruments proved challenging with the risk of falling into the jingle-jangle fallacy. The importance of meticulous examination and comparison of measurement tools became evident. Further, our review revealed a predominant research focus on men in the existing literature. In contrast, instruments targeting women and bisexual individuals, are relatively rare.

Methods

1. More information regarding the independent reviewers for the stages of screening would be helpful – are their any parts of their background (e.g., academic, personal) that would influence screening? For example, anything regarding reflexivity/positionality

Thank you for your feedback. We agree that more information is helpful for our readers given the topic of this review. We included a positionality statement at the end of the document to not overload the methods section (l. 711-720):

Positionality statement

All independent reviewers involved in screening and data extraction are researchers trained in evidence synthesis with disciplinary backgrounds in psychology, public health, and epidemiology, working within Western European public institutions. We approach the topic as scholarly allies rather than community representatives and do not claim lived-experience authority. We recognize that our personal background as well as our disciplinary training, institutional context, and the English/German scholarly lens can shape interpretation. We therefore employed rigorous methodology and reflected on potential normative assumptions and made the fewest possible adjustments to the included studies’ descriptions and documented any necessary clarifications transparently (see Data Analysis and table footnotes).

2. Details regarding the consensus process would also be helpful. Who resolved conflicts?

Thank you for your comment. We now detail this process further (l. 217-220):

At the second stage, conflicts on the in- and exclusion of references were resolved in consensus by thorough discussion between the involved screeners and, if no consensus could be reached, through discussion with all screeners (MMK, SC, SF, AJ, MSix), with resolution by majority vote.

3. Can the authors please provide the data extraction form?

Thank you for this suggestion. We now provide the data extraction form in Supplement 3.

Results

1. There are a series of measures (page 10-11, lines 214-238) that are listed at bullet points. It may be more appropriate to describe these in a narrative format and provide additional details regarding the constructs they address.

Thank you for your suggestion. We reworked that passage and now provide more information on the included constructs (p. 15f).

2. Reference(s) are needed to support the claim made on lines 240-243.

Thank you for mentioning the missing references. We’ve added the relevant reference to Meyer’s minority stress model which were missing before (l. 394-398):

The following instruments capture constructs that do not directly map onto Meyer’s minority stress model (11, 12, 41) but were included due to their conceptual relevance to minority stress experiences. While they do not assess distal or proximal stressors as defined by Meyer (11), they address themes such as intraminority dynamics, social support, and identity-related cognitions, which may influence or co-occur with minority stress.

3. For the measures listed as having been adapted (page 14, line 328), it would be helpful to know what the scope of the adaptation(s) were.

Thank you for your comment. We have updated this section and included a table putting original measurement instruments next to their adaptations in Supplement 4. The revised section in the manuscripts now reads (l. 500-502):

Across adapted measures, the most common modifications were item reductions and revisions of measur

---

## [Decision Letter · Decision Letter 1]

23 Jan 2026

Measurement instruments for sexual identity minority stress in adults: a scoping review

PONE-D-25-34195R1

Dear Dr. Misevic-Kallenbach,

We’re pleased to inform you that your manuscript has been judged scientifically suitable for publication and will be formally accepted for publication once it meets all outstanding technical requirements.

Kind regards,

Daniel Demant, PhD

Academic Editor

PLOS One

Additional Editor Comments (optional):

I would like to commend the authors for the thoroughness and quality of their revisions. The manuscript has been clearly strengthened across sections, with an improved theoretical framing, transparent and reproducible methods, a well-structured and focused presentation of results and a substantially enhanced discussion that situates the findings within contemporary measurement and minority stress scholarship. Reviewer 2’s most recent comments are very general in nature and largely repeat the tone of their initial review without acknowledging the extensive changes made. In contrast, Reviewer 1 provided detailed, field-specific feedback that has clearly been incorporated and has materially improved the paper. Taken together, the revisions demonstrate rigor, responsiveness and a clear contribution to the literature on sexual-identity minority stress measurement, and I fully support acceptance of the manuscript in its current form.

Reviewers' comments:

Reviewer's Responses to Questions

**Comments to the Author**

Reviewer #1: All comments have been addressed

Reviewer #2: All comments have been addressed

2. Is the manuscript technically sound, and do the data support the conclusions?

Reviewer #1: (No Response)

Reviewer #2: Partly

3. Has the statistical analysis been performed appropriately and rigorously?

Reviewer #1: (No Response)

Reviewer #2: Yes

4. Have the authors made all data underlying the findings in their manuscript fully available?

Reviewer #1: (No Response)

Reviewer #2: Yes

5. Is the manuscript presented in an intelligible fashion and written in standard English?

Reviewer #1: (No Response)

Reviewer #2: Yes

Reviewer #1: I appreciate the authors' responsiveness to the reviewers' comments. I have no further suggestions or edits.

Reviewer #2: We thank the authors for addtessing the comments given on the orevioua c9mmwnts. The abstract is not clear and needs revision.The introduction is weak and fail to entail what it should entail. The methods section is not complete and c9mprhensive. The result is good and it needs to be brief and focus on the main find8ngs. The dicussion is again inadequate and it should explain, justify and refere with up to date references .

Regards,

**Do you want your identity to be public for this peer review?** For information about this choice, including consent withdrawal, please see our Privacy Policy

Reviewer #1: No

Reviewer #2: No

---

## [Editor Report · Acceptance letter]

PONE-D-25-34195R1

PLOS One

Dear Dr. Misevic-Kallenbach,

I'm pleased to inform you that your manuscript has been deemed suitable for publication in PLOS One. Congratulations! Your manuscript is now being handed over to our production team.

Kind regards,

on behalf of

Associate Professor Daniel Demant

Academic Editor

PLOS One